# The physical activity health paradox and risk factors for cardiovascular disease: A cross-sectional compositional data analysis in the Copenhagen City Heart Study

**Melker S. Johansson**[1,2]*, **Andreas Holtermann**[1,2], **Jacob L. Marott**[3], **Eva Prescott**[3,4], **Peter Schnohr**[3], **Mette Korshøj**[1,5], **Karen Søgaard**[2,6]

1 Musculoskeletal Disorders and Physical Workload, National Research Centre for the Working Environment, Copenhagen, Denmark, 2 Department of Sports Science and Clinical Biomechanics, University of Southern Denmark, Odense, Denmark, 3 The Copenhagen City Heart Study, Bispebjerg and Frederiksberg Hospital, Frederiksberg, Denmark, 4 Department of Cardiology, Bispebjerg University Hospital, Copenhagen, Denmark, 5 Department of Occupational and Social Medicine, Holbæk Hospital, Holbæk, Denmark, 6 Department of Clinical Research, University of Southern Denmark, Odense, Denmark

* msjohansson@health.sdu.dk

**Data Availability Statement:** The data generated and analysed for this study contains potentially identifiable or sensitive information and can

## Abstract

### Background

Studies indicate that physical activity during leisure and work have opposite associations with cardiovascular disease (CVD) risk factors, referred to as the physical activity health paradox. We investigated how sedentary behaviour and physical activity types during leisure and work are associated with systolic blood pressure (SBP), waist circumference (WC), and low-density lipoprotein cholesterol (LDL-C) in an adult general population sample using compositional data analysis.

### Methods

Participants wore accelerometers for 7 days (right thigh and iliac crest; 24 h/day) and had their SBP, WC, and LDL-C measured. Accelerometer data was analysed using the software Acti4 to derive daily time spent in sedentary behaviour and physical activity types. The measure of association was quantified by reallocating time between sedentary behaviour and 1) walking, and 2) high-intensity physical activity (HIPA; sum of climbing stairs, running, cycling, and rowing), during both domains.

### Results

In total, 652 participants were included in the analyses (median wear time: 6 days, 23.8 h/day). During leisure, the results indicated that less sedentary behaviour and more walking or more HIPA was associated with lower SBP, while during work, the findings indicated an association with higher SBP. During both domains, the findings indicated that less sedentary behaviour and more HIPA was associated with a smaller WC and lower LDL-C. However,

therefore not be shared publicly (General Data Protection Regulation, European Union). However, anybody can apply for the use of data by contacting the secretariat director of the Copenhagen City Heart Study. For contact information, please see https://www.frederiksberghospital.dk/afdelinger-og-klinikker/oesterbroundersoegelsen/kontakt/Sider/default.aspx. The authors of the present study had no special privileges in accessing the data that other interested researchers would not have.

**Funding:** The Danish Heart Foundation, the Beckett Foundation, the Danish Lung Association, the IMK – Almene Fond, and Helene and Viggo Bruun's Foundation funded the fifth examination of the CCHS. The funders were not involved in the design and management of the study, in the collection, analysis or the interpretation of data, in the preparation of the manuscript, or in the decision to submit the manuscript for publication. MSJ received funding from the Faculty of Health Sciences, University of Southern Denmark, Odense, Denmark (Faculty Scholarship), and the National Research Centre for the Working Environment, Copenhagen, Denmark (internal funding). The remaining authors received no specific funding for this work.

**Competing interests:** The authors have declared that no competing interests exist.

the findings indicated less sedentary behaviour and more walking to be associated with a larger WC and higher LDL-C, regardless of domain.

## Conclusions

During leisure, less sedentary behaviour and more walking or HIPA seems to be associated with a lower SBP, but, during work, it seems to be associated with a higher SBP. No consistent differences between domains were observed for WC and LDL-C. These findings highlight the importance of considering the physical activity health paradox, at least for some risk factors for CVD.

## Introduction

The favourable effects of leisure time physical activity on the risk of cardiovascular disease (CVD) and mortality are well established [1–7]. In contrast, occupational physical activity may increase the risk of both CVD-specific and all-cause mortality, at least among men [8–10], and evidence on the association between occupational physical activity and risk factors for CVD, risk of ischemic heart disease (IHD), and major cardiovascular events is inconclusive [9–14]. The contrasting health effects from physical activity during leisure and work have been referred to as the *physical activity health paradox* [15].

Current physical activity recommendations are mainly based on evidence from leisure time physical activity [16,17]. However, a large proportion of the general population accumulates most of their daily physical activity *at work*, in particular groups with lower socioeconomic status [18]. Therefore, it is important to investigate the opposing health effects from occupational physical activity. The physical activity health paradox may be explained by differences in characteristics (e.g., duration, intensity, and time for restitution) and physiological responses (e.g., average 24-hour heart rate and blood pressure) of physical activity during leisure and work [19]. It has also been suggested to be explained by methodological limitations [20]. Firstly, the detrimental health effects of occupational physical activity may be confounded by socioeconomic status [20], because a low socioeconomic status is associated with high occupational physical activity [18] and poor health [21,22]. Secondly, the findings may be attributed to the use of self-reported measurements of physical activity, which compared to device-based measurements, have a higher risk of misclassification that can lead to inaccurate exposure measurements [23]. Thirdly, most previous studies have investigated associations between physical activity and risk factors for CVD [1,3,8,24] without taking the *co-dependency* between durations of different types of physical activity into account. This has both conceptual and statistical limitations that can be addressed by compositional data analysis (CoDA) [25–28].

Our study objectives were to investigate how sedentary behaviour, walking, and high intensity physical activity (HIPA) during leisure and work are associated with risk factors for CVD (i.e., SBP, WC, and LDL-C) in a general population sample using CoDA.

## Methods

### Data source and study design

For this cross-sectional study, we used data from the fifth examination of the Copenhagen City Heart Study (CCHS), collected from October 2011 to February 2015. In total, 9215 individuals who were ≥20 years old and lived in two parts of Copenhagen, Denmark, were invited of

which 4543 participated (49.3%) (Fig 1). These were randomly chosen from the Copenhagen Population Register using a national registration number. Briefly, invitations were sent three weeks prior to a planned health examination. The invitations included a questionnaire and a pre-paid postcard where the individuals could confirm, change the appointment, or decline to participate. The source population, recruitment and invitation procedure, data collection, and data processing in the CCHS are described in detail elsewhere [29,30]. We followed the Strengthening the Reporting of Observational Studies in Epidemiology (STROBE) Statement to report this study.

The Danish Data Protection Agency approved the analysis of the study data (approval no.: 2001-54-0280; 2007-58-0015, 2012-58-0004, HEH-2015-045, I-suite 03741). The National Committee on Health Research Ethics approved the data collection (approval no.: VEK: H-KF 01-144/01 31104). Participation was voluntary and in agreement with the Declaration of Helsinki. Written consent to participate in the fifth examination of the CCHS was obtained from the participants.

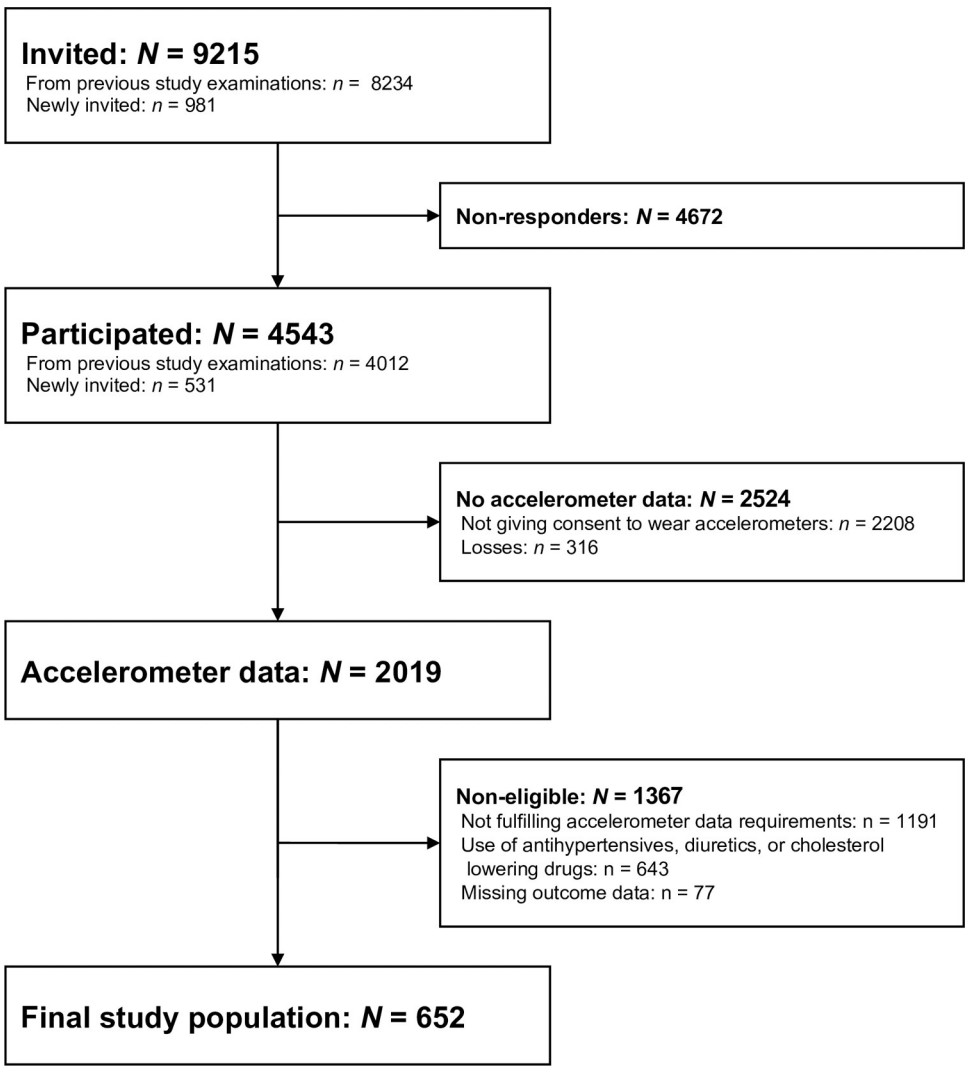

**Fig 1. Formation of the final study population of eligible participants in the fifth examination of the Copenhagen City Heart Study.** *N/n* indicates number of participants. The sum across reasons for exclusion of non-eligible study participants exceeds 1367 since some participants fulfilled more than one exclusion criterion.

## Data collection

**Questionnaire.**   Data across a wide range of domains were collected with a self-administered questionnaire. See S1 Table for an overview of the questions relevant for this study.

**Physical examination.**   The study participants underwent a physical examination at a public hospital in the Capital Region of Denmark. Medical specialists, medical students and medical laboratory technicians who were trained in the examination procedures undertook the examination.

For the current study, the relevant tests were measurements of blood pressure, WC, and LDL-C. We also used height and weight for descriptive purposes. Three blood pressure measurements were taken on participants' non-dominant arm using an automatic blood pressure monitor (OMRON M3, OMRON Healthcare, Hoofddorp, Netherlands) after five minutes of rest in a seated position. This test procedure has been used in previous examinations of the CCHS and is in line with the 2020 International Society of Hypertension Global Hypertension Practice Guidelines [31]. The participants' WC was measured at the estimated midpoint between the lower margin of the last palpable rib and the top of the iliac crest. Using standardised procedures, venipunctures were taken, and LDL-C was determined directly (Sanofi Genzyme, Cambridge, Massachusetts, USA). Height was measured without shoes on a fixed scale to the nearest 0.1 centimetre. Weight was measured with clothes but without shoes, on a consultation scale (Seca, Hamburg, Germany) to the nearest 0.1 kilogram.

**Accelerometer-based measurement of physical activity.**   As part of a sub-study, all participants were asked to wear accelerometers 24-hours per day for seven consecutive days to measure their physical activity. Totally, 2335 participants gave consent and had one tri-axial accelerometer attached on the anterior aspect of the right thigh midway between the greater trochanter and patella oriented along the axis of the thigh, and a second accelerometer on the lateral aspect of the right iliac crest (ActiGraph GT3X+; sampling frequency: 30 Hz; ActiGraph, Pensacola, Florida, USA). To secure a fixed position during the measurement period, we attached the accelerometers to the skin using double-sided medical tape (Hair-Set for hairpieces; 3M, Maplewood, Minnesota, USA) and covered them with adhesive film (OpSite Flexifix; Smith & Nephew, London, UK).

The study staff asked the study participants to fill out a diary with their working hours, leisure time, time in bed, and periods of non-wear time, and to make a daily reference measurement by standing still for 15 seconds and note the time in the diary. In addition, the participants were asked to only remove the accelerometers if they visited a sauna or in case of adverse skin reactions, discomfort, pain, or affected sleep. Finally, the participants were asked to return the accelerometers at the test centre or by mail using a pre-paid envelope after the measurement period. The accelerometers were initialised, and raw data was downloaded with the manufacturer's software (ActiLife version 5) by the study staff.

## Processing of raw accelerometer data

**Detection of body postures and physical activity types.**   We used the MATLAB-software Acti4 (National Research Centre for the Working Environment, Copenhagen, Denmark) to detect and derive the daily time spent lying, sitting, standing, moving (i.e., small movements without regular walking while in a standing posture), walking, climbing stairs (i.e., up and down), running, cycling and rowing. The detection of body postures and physical activity types is based on an algorithm that uses inclinations and accelerations from the accelerometers [32]. The sensitivity and specificity of this activity-classification has been reported to be >90% during standardised and semi-standardised conditions for all body postures and physical activity types except climbing stairs which has a lower sensitivity (i.e., sensitivity: 75.4%; specificity:

99.7%) [32,33]. We used the daily reference measurements to define the individual angle between the vertical axis of the accelerometer and the axis of the thigh, which was used in the activity-detection algorithm.

Leisure and work, time in bed, quality control, and non-wear time. We defined each participant's leisure time and working hours based on diary information. This made it possible to derive time spent in the physical activity types during leisure and work, respectively.

Time in bed was defined by a combination of accelerometer and diary data (i.e., self-reported bedtime/get up time). More specifically, we adjusted any inconsistencies between the diary data and detected lying/non-lying activity types of more than 15 minutes, by setting the time to the nearest five minutes of the observed lying/non-lying activity.

As a quality control, we visually inspected the activity classification over time for each individual, to identify and investigate any abnormalities in the data (e.g., extreme levels of rowing or lack of sitting).

We manually added non-wear time based on diary information and visual inspection of the activity-classification over time. Additionally, Acti4 detects non-wear time automatically based on criteria that has been described in detail elsewhere [32].

## Eligibility criteria

We included participants who had registered work time during the measurement period and ≥5 days of measurements with ≥16 h of accelerometer recordings per day. There were no requirements on number of workdays, number of hours of work per day, or day of the week (i.e., weekday or weekend day). Any days marked as 'sick days' in the diary were excluded. Participants who reported use of antihypertensive, diuretics or cholesterol lowering drugs, or had missing values in any of the outcome variables were excluded.

## Definition of variables

**Physical activity composition.**   The daily physical activity composition consisted of time spent *sedentary* (i.e., sum of lying and sitting), *standing*, *moving*, *walking*, and in *HIPA* (i.e., sum of climbing stairs, running, cycling, and rowing) during leisure and work, plus *time in bed*. Since not all participants climbed stairs, ran, cycled, or rowed during the measurement period, some participants had zero minutes in these activity types. Because physical activity types (i.e., compositional parts) that consist of zeros cannot be included in CoDA, we merged climbing stairs, running, cycling and rowing into the combined activity class HIPA.

**Outcomes.**   As outcome variables, we used SBP (mm Hg), WC (cm), and LDL-C (mmol/L). We used the average of the three blood pressure measurements.

**Covariates and variables for descriptive analyses.**   Sex, age, number of years of education, smoking status, average number of units of alcohol per week, and self-reported use of prescribed medication for cardiovascular disease, antidepressants or sedatives, asthma or bronchitis, or diabetes were included as covariates in our analyses.

For descriptive purposes, we categorised body mass index (BMI, calculated as weight in kilograms divided by height in meters squared) into *underweight* ($<18.5$ kg/m$^2$), *normal weight* ($18.5$-$<25.0$ kg/m$^2$), *overweight* ($25.0$-$<30.0$ kg/m$^2$), and *obese* ($\geq30$ kg/m$^2$) [34]. Furthermore, we categorised blood pressure into *normal* (systolic: $<140$ mm Hg and diastolic: $<90$ mm Hg; i.e., includes high normal), *grade 1 hypertension* (systolic: $140$-$\leq159$ mm Hg or diastolic: $90$-$\leq99$ mm Hg), *grade 2 hypertension* (systolic: $160$-$\leq179$ mm Hg or diastolic: $100$-$\leq109$ mm Hg), and *grade 3 hypertension* (systolic: $\geq180$ mm Hg or diastolic: $\geq110$ mm Hg) [35]. Finally, WC was categorised into $>88$ cm for women and $>94$ cm for men [36]. Further details about how these variables were derived can be found in S2 Table.

## Statistical analyses

**Descriptive statistics.**   We used frequencies with percentages or medians with the first and third quartile (Q1-Q3) to describe the characteristics of the study population. Medians were used due to skewed distributions of some of the continuous variables.

**Investigation of selection bias.**   We compared the characteristics of the study participants who did *not* fulfil the inclusion criteria with those who fulfilled using Mann-Whitney U test, Pearson's Chi-squared test (p-values <0.05 were considered to indicate differences between groups) and by assessing 95% confidence intervals (CI) of proportions and medians. The CIs were calculated with the Wilson's score method [37] and the normal approximation method for proportions and medians, respectively.

**Data transformation.**   The sample space of compositional data (i.e., the simplex) has a geometry that is incompatible with standard statistical methods. To make these methods applicable, we transformed the physical activity composition with the *isometric log-ratio* (ilr) transformation [25,38]. This resulted in a set of *ilr*-coordinates that represent the physical activity composition in a sample space (i.e., the *real* coordinate space) that allows the use of standard statistical methods [26]. Specifically, we constructed *pivot ilr*-coordinates, in which the first coordinate (*ilr1*) represents the first part of the composition relative to the geometric mean of the remaining parts [38].

**Modelling process and reallocation of time.**   We investigated how the physical activity composition (expressed as *ilr*-coordinates) were associated with each outcome using linear regression models (i.e., crude and adjusted analyses). The modelling process was conducted through three steps:

i. Firstly, we fitted multiple linear regression models with the *ilr*-coordinates representing the physical activity composition and potential confounders as covariates (i.e., only in the adjusted analyses) and SBP, WC, and LDL-C as outcome. Observations with missing values in the covariates were not included in the adjusted models (n = 69). The model assumptions were checked by plotting standardised residuals against a) continuous covariates (i.e., assumption of linearity) and b) fitted values (i.e., assumption of homogeneous variance), and by quantile-quantile (Q-Q) plots of the residuals (i.e., assumption of normally distributed residuals).

ii. Secondly, because the model estimates of the *ilr*-coordinates are not directly interpretable (due to the *ilr*-transformation), we theoretically reallocated time between sedentary behaviour and 1) walking, and 2) HIPA to quantify the measure of association in an understandable way [26]. Specifically, for work and leisure, respectively, we reallocated the geometric mean composition (i.e., reference composition) according to time reallocation 1 and 2. That is, the reallocations were made pairwise (a.k.a. *one-to-one reallocations*) during work and during leisure, respectively; all remaining physical activity types were kept constant. For example, if 10 minutes were reallocated *from* sedentary behaviour *to* walking in a theoretical composition consisting of 315 minutes sedentary behaviour, 100 minutes standing, 60 minutes walking and 5 minutes HIPA *during work*, it would result in 305 minutes sedentary behaviour and 70 minutes walking during work, with the duration of the remaining physical activity types, in both domains, kept constant.
Because the geometric mean of walking and HIPA was lower during work than leisure, we could not reallocate the same absolute duration of time during work and leisure. For time reallocation 1, we therefore reallocated 10 to 30 minutes between sedentary behaviour and walking during work, and 10 to 50 min during leisure time, in 10-minute portions. For

time reallocation 2, we reallocated 1 to 2 minutes between sedentary behaviour and HIPA during work, and 1 to 10 minutes during leisure, in 1- and 2-minute portions.

iii. Thirdly, we used the fitted values from the linear regression models to estimate each outcome given the reference- and reallocated compositions. Then, we calculated the *difference* in outcome by subtracting the estimated outcome of the reference-composition from the estimated outcome of each reallocated composition [26,27].

**Sensitivity analyses.** To investigate the influence of excluding individuals taking antihypertensives, diuretics, or cholesterol lowering drugs, we conducted sensitivity analyses including 1) all study participants regardless of medication use, and 2) limited to those with the medications use.

We used RStudio (version 1.3.1093) [39] running R (version 4.0.3) [40] for all analyses, and, specifically, the packages *compositions* [41] and *robCompositions* [42] for the analyses involving CoDA.

## Results

### Cohort characteristics

We have illustrated the cohort formation in Fig 1 and presented characteristics of the study population in Table 1. The median number of valid days was 6 and the study participants wore the accelerometers for a median time of 23.8 h/day. Furthermore, the median number of

**Table 1. Characteristics of 652 adults participating in the fifth examination of the Copenhagen City Heart Study.**

| N = 652 | |
|---|---|
| **Characteristics** | **n (%) / Median (Q1, Q3)** |
| Accelerometer wear time | 652 (100.0) |
| Median h/day | 23.8 (23.1, 24.0) |
| Number of valid days of measurement | 652 (100.0) |
| Median number of days | 6.0 (6.0, 7.0) |
| Working hours | 652 (100.0) |
| Median h/day | 7.6 (6.7, 8.3) |
| Number of workdays | 652 (100.0) |
| Median number of days | 4.0 (3.0, 4.0) |
| Sex distribution | 652 (100.0) |
| Women | 378 (58.0) |
| Men | 274 (42.0) |
| Age | 652 (100.0) |
| Median years | 48.6 (36.1, 57.1) |
| Number of years of education | 652 (100.0) |
| Median years | 13.0 (12.0, 14.0) |
| Level of education | 651 (99.9) |
| No further education beyond primary school | 47 (7.2) |
| Short education (up to 3 years) | 44 (6.8) |
| Vocational or comparable education (1–3 years) | 105 (16.1) |
| Higher education (≥3 years) | 176 (27.0) |
| University education | 279 (42.9) |
| Household income | 644 (98.8) |

(*Continued*)

**Table 1.** (Continued)

| *N* = 652 | |
|---|---|
| **Characteristics** | **n (%) / Median (Q1, Q3)** |
| Low (<200 000 DKK) | 69 (10.7) |
| Middle (200 000–600 000 DKK) | 238 (37.0) |
| High (≥600 000 DKK) | 337 (52.3) |
| Smoking status | 639 (98.0) |
| Non-smoker | 295 (46.2) |
| Previous smoker | 253 (39.6) |
| Current smoker | 91 (14.2) |
| Average weekly number of units of alcohol per week | 594 (91.1) |
| Median units/week | 6.0 (3.0, 11.0) |
| Use of prescribed medication | 652 (100.0) |
| Yes | 49 (7.5) |
| Self-reported general health | 648 |
| Excellent or Very good | 328 (50.6) |
| Good | 256 (39.5) |
| Less good or Poor | 64 (9.9) |
| Systolic blood pressure | 652 (100.0) |
| Median (mmHg) | 127.8 (117.5, 138.0) |
| Diastolic blood pressure | 652 (100.0) |
| Median (mmHg) | 77.0 (70.5, 83.5) |
| Blood pressure classification | 652 (100.0) |
| Normal | 486 (74.5) |
| Grade 1 hypertension | 147 (22.6) |
| Grade 2 or 3 hypertension | 19 (2.9) |
| Waist circumference | 652 (100.0) |
| Median (cm) | 83.0 (76.0, 92.0) |
| Waist circumference classification | 652 (100.0) |
| Women >80 cm | 413 (63.3) |
| Men >94 cm | 239 (36.7) |
| BMI | 652 (100.0) |
| Underweight | 7 (1.1) |
| Normal | 393 (60.3) |
| Overweight | 203 (31.1) |
| Obese | 56 (8.6) |
| Low density lipoprotein cholesterol | 652 (100.0) |
| Median (mmol/L) | 3.0 (2.5, 3.7) |

N/n, number of observations.

y, years.

Q1-Q3, first and third quartile.

DKK, Danish kroner.

mm Hg, millimetre of mercury.

Blood pressure classification is based on the 2013 European Society of Hypertension/European Society of Cardiology guidelines for the management of arterial hypertension (the normal category includes high normal).

BMI, body mass index.

BMI was classified according to the WHO classification (Underweight, <18.5 kg/m$^2$; Normal weight, 18.5-<25.0 kg/m$^2$; Overweight, 25.0-<30.0 kg/m$^2$; Obese, ≥30 kg/m$^2$.

mmol/L, millimol per litre (molar concentration).

workdays was 4 and 94% had >1 workday. The median worktime was 7.6 h/day. There were 58% women, and the median age was 48.6 years. The median SBP, WC, and LDL-C was 128 mm Hg, 83 cm, and 3.0 mmol/L, respectively.

The geometric mean of each part of the physical activity composition is presented in Table 2, stratified by leisure and work.

## Investigation of selection bias

The study participants who did *not* fulfil the inclusion criteria were older, had a lower level of education, and a lower household income, a higher median SBP and WC, a lower LDL-C, and a higher proportion were previous or current smokers, reported use of prescribed medicine, rated their general health as less good or poor, and were classified as overweight and obese, and with hypertension compared to those who fulfilled the inclusion criteria (S4 Table).

## Model validation

The model validation did not reveal any substantial violations of the model assumptions (S1 File). However, specifically for the SBP-model, the variation of the standardised residuals slightly increased across the fitted values. For all three models, the residuals were not perfectly normally distributed, but the deviations were considered too small to substantially affect the model fit.

## Time reallocations

All estimates presented here are from the adjusted analyses. Results from the crude analyses can be found in S2 File. We emphasise that all time reallocations were made with the mean composition as the starting point.

**Systolic blood pressure.** During leisure, the results indicated that less sedentary behaviour and more walking compared to the reference composition was associated with a *lower* SBP, while the results indicated an association with a *higher* SBP during work (Fig 2A and Table 3). Importantly, the size of the estimated differences in SBP differed markedly between the domains. For example, the absolute difference in SBP given 30 minutes less walking and 30 minutes more sedentary behaviour during work was 11 times larger than that during leisure (work: -6.7 [95% CI: -16.2, 2–9] mm Hg vs. leisure: 0.6 [-2.7, 3.8] mm Hg). The same pattern of opposite associations was evident for less sedentary behaviour and more HIPA during leisure and work. Although the CIs included zero, the majority of the values indicated a lower

**Table 2. Geometric mean of 24-h physical activity composition among 652 participants in the fifth examination of the Copenhagen City Heart Study stratified by domain.**

| Physical activity type | Domain | |
|---|---|---|
| | **Leisure**<br>**Minutes (%) of a 24-h day** | **Work**<br>**Minutes (%) of 24-h day** |
| Sedentary behaviour | 372.71 (25.88) | 234.24 (16.27) |
| Standing | 126.97 (8.82) | 74.62 (5.18) |
| Moving | 49.10 (3.41) | 22.46 (1.56) |
| Walking | 56.80 (3.94) | 32.51 (2.26) |
| HIPA | 10.88 (0.76) | 2.46 (0.17) |
| Time in bed | 457.26 (31.75) | - |

HIPA, high-intensity physical activity (sum of climbing stairs, running, cycling and rowing).

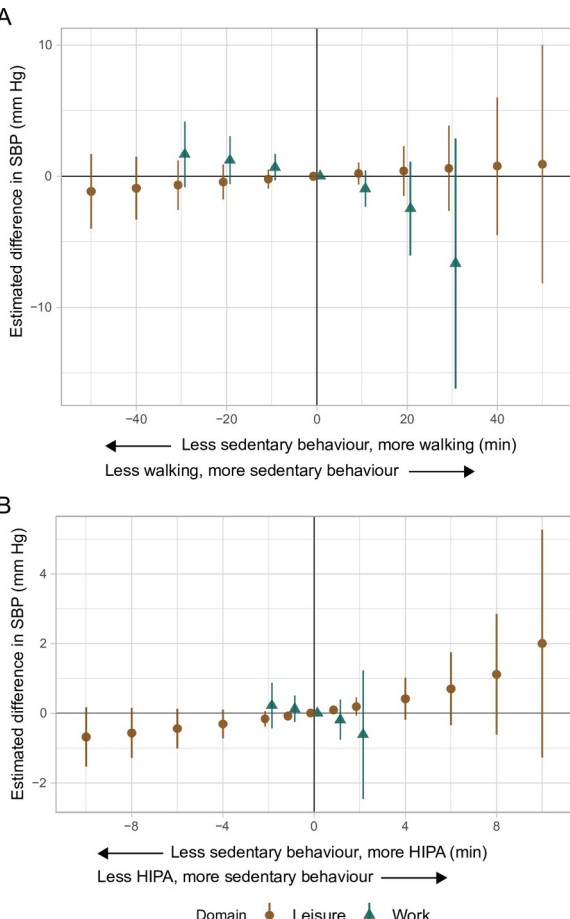

**Fig 2.** Adjusted estimated differences in systolic blood pressure (mm Hg, y-axis) given the reallocation of time between sedentary behaviour and **A)** walking, and **B)** HIPA among 652 adults. A negative value on the x-axis reflects the pairwise reallocation of time *from* sedentary behaviour *to* walking or HIPA, while a positive value reflects the reallocation of time *from* walking or HIPA *to* sedentary behaviour. The origin represents the reference composition (i.e., 372.7 and 234.2 min sedentary behaviour, 127.0 and 74.6 min standing, 49.1 and 22.5 min moving, 56.8 and 32.5 min walking, and 10.9 and 2.5 min HIPA, during leisure and work, respectively, and 457.3 min in bed). The difference in outcome was calculated by subtracting the estimated outcome of the reference composition from the estimated outcome for each reallocated composition. SBP is systolic blood pressure. HIPA is high-intensity physical activity (i.e., sum of climbing stairs, running, cycling, and rowing). Vertical lines correspond to the 95% confidence intervals.

and higher SBP during leisure and work, respectively (e.g., 10 min, leisure: -0.7, 95% CI: -1.5, 0.2; Fig 2B and Table 4).

**Waist circumference.** During both leisure and work, the results indicated less sedentary behaviour and more walking to be associated with a larger WC; however, the CIs included zero (Fig 3A and Table 3). In contrast, during leisure and work, less sedentary behaviour and more HIPA was associated with a smaller WC, although the estimates during work were small. Also, for work, the CIs included zero, but most values suggested a smaller WC (Fig 3B and Table 4). The estimated difference in WC given the time reallocations was not symmetric. For example, during work, the reallocation of 30 min walking to sedentary behaviour was associated with a 5 cm smaller WC (95% CI: -11.29, 1.03) compared to an estimated 1 cm larger WC given the opposite time reallocation. Additionally, the smaller WC (i.e., -5 cm) is about five times larger than the estimated difference observed for the corresponding time reallocation during leisure (i.e., -1 cm).

**Table 3. Estimated adjusted differences in systolic blood pressure, waist circumference, and low-density lipoprotein cholesterol given time reallocations between sedentary behaviour and walking during leisure and work among 652 adults in the fifth examination of the Copenhagen City Heart Study.**

| Time reallocations | Leisure Estimated difference in outcome (95% CI) | Work Estimated difference in outcome (95% CI) |
|---|---|---|
| **Systolic blood pressure (mm Hg)** | | |
| -50 min (sedentary behaviour → walking) | -1.16 (-4.01, 1.69) | - |
| -40 | -0.91 (-3.30, 1.48) | - |
| -30 | -0.67 (-2.56, 1.22) | 1.67 (-0.83, 4.16) |
| -20 | -0.44 (-1.78, 0.90) | 1.23 (-0.60, 3.05) |
| -10 | -0.22 (-0.93, 0.50) | 0.69 (-0.33, 1.70) |
| 0 (reference composition) | 0 (0, 0) | 0 (0, 0) |
| 10 | 0.21 (-0.64, 1.06) | -0.95 (-2.33, 0.43) |
| 20 | 0.41 (-1.48, 2.30) | -2.47 (-6.04, 1.10) |
| 30 | 0.60 (-2.66, 3.85) | -6.66 (-16.19, 2.88) |
| 40 | 0.77 (-4.47, 6.02) | - |
| 50 min (walking → sedentary behaviour) | 0.92 (-8.17, 10.00) | - |
| **Waist circumference (cm)** | | |
| -50 min (sedentary behaviour → walking) | 0.56 (-1.29, 2.40) | - |
| -40 | 0.50 (-1.05, 2.05) | - |
| -30 | 0.42 (-0.80, 1.65) | 1.25 (-0.37, 2.86) |
| -20 | 0.32 (-0.55, 1.18) | 0.92 (-0.26, 2.10) |
| -10 | 0.18 (-0.28, 0.64) | 0.52 (-0.14, 1.18) |
| 0 (reference composition) | 0 (0, 0) | 0 (0, 0) |
| 10 | -0.24 (-0.79, 0.31) | -0.72 (-1.62, 0.17) |
| 20 | -0.56 (-1.78, 0.66) | -1.89 (-4.20, 0.42) |
| 30 | -1.03 (-3.13, 1.08) | -5.13 (-11.29, 1.03) |
| 40 | -1.75 (-5.14, 1.64) | - |
| 50 min (walking → sedentary behaviour) | -3.26 (-9.13, 2.62) | - |
| **Low-density lipoprotein cholesterol (mmol/L)** | | |
| -50 min (sedentary behaviour → walking) | 0.16 (-0.01, 0.33) | - |
| -40 | 0.14 (-0.003, 0.28) | - |
| -30 | 0.11 (-0.001, 0.22) | 0.09 (-0.05, 0.24) |
| -20 | 0.08 (0.001, 0.160) | 0.07 (-0.04, 0.18) |
| -10 | 0.04 (0.001, 0.090) | 0.04 (-0.02, 0.10) |
| 0 (reference composition) | 0 (0, 0) | 0 (0, 0) |
| 10 | -0.05 (-0.10, -0.003) | -0.05 (-0.13, 0.03) |
| 20 | -0.12 (-0.23, -0.01) | -0.13 (-0.35, 0.08) |
| 30 | -0.21 (-0.41, -0.02) | -0.35 (-0.92, 0.21) |
| 40 | -0.35 (-0.66, -0.04) | - |

(*Continued*)

**Table 3.** (Continued)

| Time reallocations | Leisure Estimated difference in outcome (95% CI) | Work Estimated difference in outcome (95% CI) |
|---|---|---|
| 50 min (walking → sedentary behaviour) | -0.62 (-1.16, -0.08) | - |

Analyses adjusted for age, sex, level of education, number of alcohol units/week, smoking status, and use of prescribed medication.

69 observations were not included in the adjusted models due to missing values in some covariates.

CI, confidence interval.

mm Hg, mm of mercury.

mmol/L, mmol per litre.

Reference composition corresponds to: 372.7 and 234.2 min sedentary behaviour, 127.0 and 74.6 min standing, 49.1 and 22.5 min moving, 56.8 and 32.5 min walking, and 10.9 and 2.5 min HIPA, during leisure and work, respectively, and 457.3 min in bed (i.e., geometric mean).

HIPA, high-intensity physical activity which consists of climbing stairs (up/down), running, cycling, and rowing.

**Low-density lipoprotein cholesterol.** During both leisure and work, the results indicated that less sedentary behaviour and more walking was associated with a higher LDL-C (e.g., 20 min: 0.08, 95% CI: 0.00, 0.16 mmol/L) (Fig 4A and Table 3). During leisure, less sedentary behaviour and more HIPA was associated with a lower LDL-C (e.g., 10 min: -0.07, 95% CI: -0.12, -0.02 mmol/L). During work, the estimates followed the same pattern but were smaller and the CIs included zero (Fig 4B and Table 4).

**Table 4. Estimated adjusted differences in systolic blood pressure, waist circumference, and low-density lipoprotein cholesterol given time reallocations between sedentary behaviour and high intensity physical activity during leisure and work among 652 adults in the fifth examination of the Copenhagen City Heart Study.**

| Time reallocations | Leisure Estimated difference in outcome (95% CI) | Work Estimated difference in outcome (95% CI) |
|---|---|---|
| **Systolic blood pressure (mm Hg)** | | |
| -10 min (sedentary behaviour → HIPA) | -0.69 (-1.54, 0.17) | - |
| -8 | -0.57 (-1.29, 0.15) | - |
| -6 | -0.44 (-1.02, 0.13) | - |
| -4 | -0.31 (-0.72, 0.10) | - |
| -2 | -0.16 (-0.38, 0.06) | 0.22 (-0.44, 0.88) |
| -1 | -0.08 (-0.20, 0.03) | 0.13 (-0.25, 0.50) |
| 0 (reference composition) | 0 (0, 0) | 0 (0, 0) |
| 1 | 0.09 (-0.04, 0.22) | -0.19 (-0.77, 0.38) |
| 2 | 0.19 (-0.08, 0.45) | -0.62 (-2.47, 1.23) |
| 4 | 0.41 (-0.18, 1.01) | - |
| 6 | 0.70 (-0.34, 1.74) | - |
| 8 | 1.11 (-0.61, 2.84) | - |
| 10 min (HIPA → sedentary behaviour) | 2.00 (-1.28, 5.26) | - |
| **Waist circumference (cm)** | | |
| -10 min (sedentary behaviour → HIPA) | -1.35 (-1.90, -0.80) | - |
| -8 | -1.14 (-1.60, -0.67) | - |
| -6 | -0.90 (-1.27, -0.53) | - |

(*Continued*)

**Table 4.** (Continued)

| Time reallocations | Leisure Estimated difference in outcome (95% CI) | Work Estimated difference in outcome (95% CI) |
|---|---|---|
| -4 | -0.64 (-0.90, -0.38) | - |
| -2 | -0.34 (-0.49, -0.20) | -0.18 (-0.61, 0.25) |
| -1 | -0.18 (-0.25, -0.11) | -0.10 (-0.35, 0.14) |
| 0 (reference composition) | 0 (0, 0) | 0 (0, 0) |
| 1 | 0.20 (0.11, 0.28) | 0.16 (-0.22, 0.53) |
| 2 | 0.41 (0.24, 0.58) | 0.50 (-0.70, 1.69) |
| 4 | 0.92 (0.53, 1.30) | - |
| 6 | 1.60 (0.92, 2.27) | - |
| 8 | 2.63 (1.51, 3.74) | - |
| 10 min (HIPA → sedentary behaviour) | 4.92 (2.80, 7.03) | - |
| **Low-density lipoprotein cholesterol (mmol/L)** | | |
| -10 min (sedentary behaviour → HIPA) | -0.07 (-0.12, -0.02) | - |
| -8 | -0.06 (-0.11, -0.02) | - |
| -6 | -0.05 (-0.08, -0.02) | - |
| -4 | -0.04 (-0.06, -0.01) | - |
| -2 | -0.02 (-0.03, -0.01) | -0.01 (-0.05, 0.03) |
| -1 | -0.01 (-0.020, -0.003) | -0.01 (-0.03, 0.02) |
| 0 (reference composition) | 0 (0, 0) | 0 (0, 0) |
| 1 | 0.01 (0.003, 0.018) | 0.01 (-0.02, 0.04) |
| 2 | 0.02 (0.01, 0.04) | 0.03 (-0.08, 0.14) |
| 4 | 0.05 (0.02, 0.09) | - |
| 6 | 0.09 (0.03, 0.15) | - |
| 8 | 0.14 (0.04, 0.25) | - |
| 10 min (HIPA → sedentary behaviour) | 0.27 (0.07, 0.46) | - |

Model adjusted for age, sex, level of education, number of alcohol units/week, smoking status, and use of prescribed medication.

69 observations were not included in the adjusted models due to missing values in some covariates.

CI, confidence interval.

mm Hg, mm of mercury.

mmol/L, mmol per litre.

Reference composition corresponds to: 372.7 and 234.2 min sedentary behaviour, 127.0 and 74.6 min standing, 49.1 and 22.5 min moving, 56.8 and 32.5 min walking, and 10.9 and 2.5 min HIPA, during leisure and work, respectively, and 457.3 min in bed (i.e., geometric mean).

HIPA, high-intensity physical activity which consists of climbing stairs (up/down), running, cycling, and rowing.

**Sensitivity analyses.** Similar results were observed across the three outcomes when study participants taking antihypertensives, diuretics, or cholesterol lowering drugs were included in the analyses (Table A-C in S3 File). When the analyses were limited to those taking these drugs (n = 146), the estimated differences in SBP for time reallocations between sedentary behaviour and walking followed the same pattern but were larger than in the main analyses. However, the estimated differences in SBP given time reallocations between sedentary behaviour and HIPA followed an opposite pattern compared to the main analysis (Table D in S3 File).

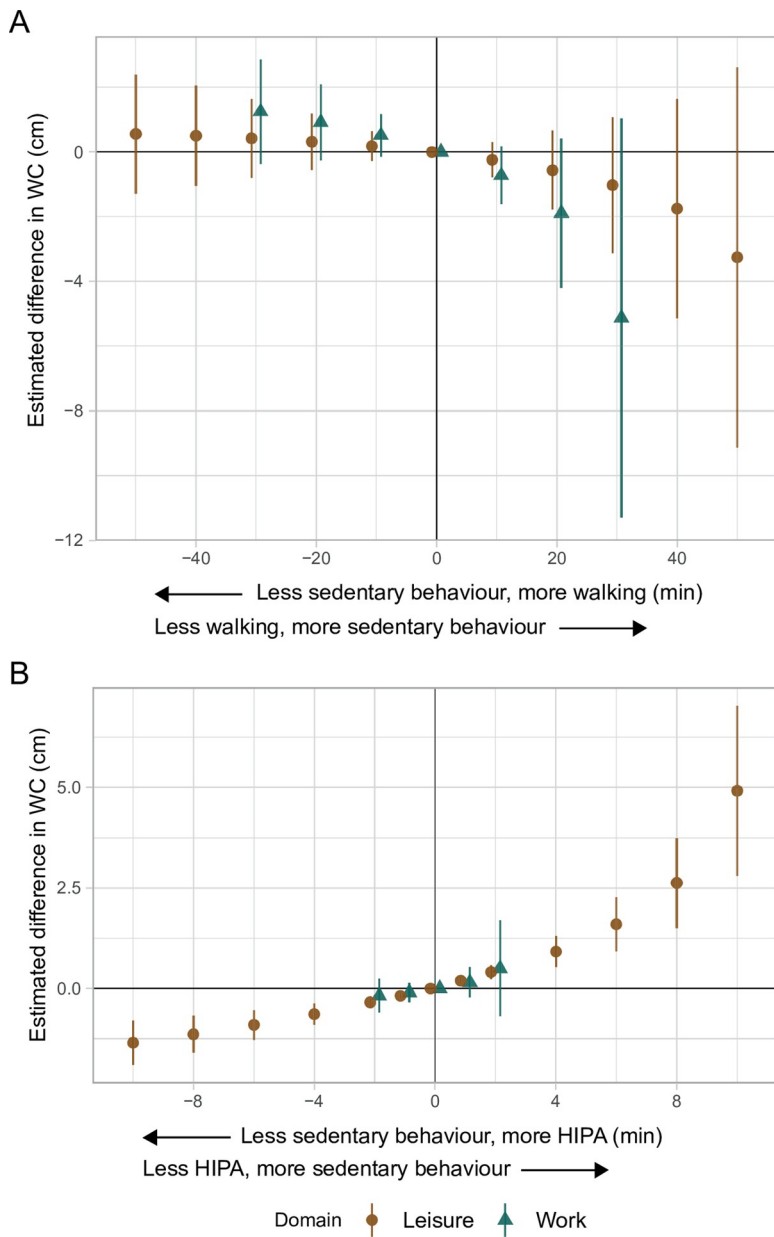

**Fig 3.** Adjusted estimated differences in waist circumference (cm, y-axis) given the reallocation of time between sedentary behaviour and **A)** walking, and **B)** HIPA among 652 adults. A negative value on the x-axis reflects the pairwise reallocation of time *from* sedentary behaviour *to* walking or HIPA, while a positive value reflects the reallocation of time *from* walking or HIPA *to* sedentary behaviour. The origin represents the reference composition (i.e., 372.7 and 234.2 min sedentary behaviour, 127.0 and 74.6 min standing, 49.1 and 22.5 min moving, 56.8 and 32.5 min walking, and 10.9 and 2.5 min HIPA, during leisure and work, respectively, and 457.3 min in bed). The difference in outcome was calculated by subtracting the estimated outcome of the reference composition from the estimated outcome for each reallocated composition. WC is waist circumference. HIPA is high-intensity physical activity (i.e., sum of climbing stairs, running, cycling, and rowing). Vertical lines correspond to the 95% confidence intervals.

Opposite patterns were also found for WC and LDL-C. Specifically, for WC in the sedentary behaviour and walk-reallocations during leisure and the sedentary behaviour and HIPA-reallocations during work, and for LDL-C in the sedentary behaviour and walk-reallocations

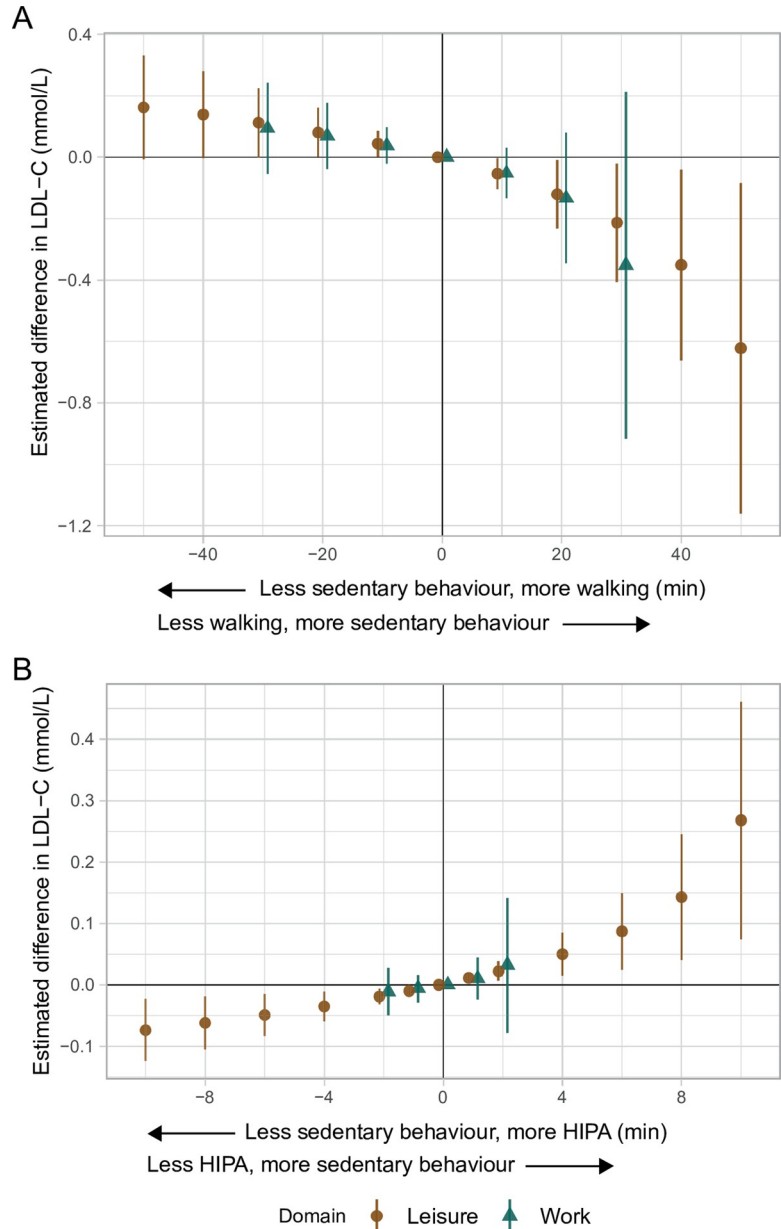

**Fig 4.** Adjusted estimated differences in low-density lipoprotein cholesterol (mmol/L, y-axis) given the reallocation of time between sedentary behaviour and **A)** walking, and **B)** HIPA among 652 adults. A negative value on the x-axis reflects the pairwise reallocation of time *from* sedentary behaviour *to* walking or HIPA, while a positive value reflects the reallocation of time *from* walking or HIPA *to* sedentary behaviour. The origin represents the reference composition (i.e., 372.7 and 234.2 min sedentary behaviour, 127.0 and 74.6 min standing, 49.1 and 22.5 min moving, 56.8 and 32.5 min walking, and 10.9 and 2.5 min HIPA, during leisure and work, respectively, and 457.3 min in bed). The difference in outcome was calculated by subtracting the estimated outcome of the reference composition from the estimated outcome for each reallocated composition. LDL-C is low-density lipoprotein cholesterol. HIPA is high-intensity physical activity (i.e., sum of climbing stairs, running, cycling, and rowing). Vertical lines correspond to the 95% confidence intervals.

during both domains and in the sedentary behaviour and HIPA-reallocations during work (Table E and F in S3 File).

## Discussion

### Summary of findings

During leisure, the findings indicated less sedentary behaviour and more walking or more HIPA to be associated with a lower SBP, while during work, the findings indicated an association with a higher SBP. During both domains, the findings indicated that less sedentary behaviour and more HIPA was associated with a smaller WC and a lower LDL-C. Furthermore, the findings indicated less sedentary behaviour and more walking to be associated with a larger WC and a higher LDL-C, regardless of domain.

### Interpretation of findings

**Systolic blood pressure.**    During leisure, the results indicated less sedentary behaviour and more walking or more HIPA compared to the reference composition to be associated with a lower SBP. In contrast, during work, the results indicated an association with a higher SBP (Fig 2, Tables 3 and 4). Although not statistically significant, these findings support that these physical activity types can have either beneficial or detrimental associations with a CVD risk factor depending on domain [8,15]. Importantly, the results from the time reallocations should be seen relative to the reference composition (Table 2).

These findings may be explained by differences in characteristics between physical activity during leisure and work [19]. Regular physical activity of moderate or higher intensity that takes place during relatively short time periods may, given sufficient time for restitution, facilitate beneficial central and peripheral adaptations of the cardiovascular system (e.g., lower heart rate, blood pressure, and inflammatory biomarkers), which decrease the risk of CVD. However, contextual factors and work conditions (e.g., productive demands, degree of control, heavy lifting, and awkward or static body postures) [15,19] make occupational physical activity different from physical activity during leisure with regards to the intensity, duration, and variation of the physical activity, as well as restitution [15,19]. Combinations of high occupational physical activity and insufficient restitution have been suggested to increase average daily heart rate, blood pressure, and levels of inflammatory biomarkers [13,15,19], which all increase the risk of CVD [19]. These mechanisms could explain our findings of a contrasting association between physical activity types during leisure and work, and SBP (Fig 2, Tables 3 and 4). As previously mentioned, measurement error (i.e., use of self-reported physical activity data) has also been suggested to explain the physical activity health paradox. Our findings do not support this since they are based on device-based measurements of physical activity.

There is considerable evidence that leisure time physical activity has favourable effects on SBP [6,43–45], while sedentary behaviour during leisure seems to be weakly associated with SBP [46]. However, to our knowledge, fewer studies have investigated how occupational physical activity are associated with SBP, and their findings are inconclusive [13,47–53]. For example, among studies investigating both leisure time and occupational physical activity, two studies found an association between higher *leisure* time physical activity and a lower SBP [13,48], which is in agreement with the findings in this study (i.e., the reallocation of time from sedentary behaviour to walking or to HIPA during leisure). In addition, one study found higher leisure time physical activity to be associated with a higher SBP [51], while two other studies did not find any association [13,50]. The results in the current study disagree with these studies. On the other hand, our findings related to the reallocation of time from sedentary behaviour to walking or HIPA during *work* (i.e., indications of an association with a higher SBP) agree with three of these studies [13,52,53], but disagree with six other studies [13,47–51]; of which four did not find any association [13,49–51]. Only two of these previous

studies used accelerometer data [13,48], and only one used CoDA [48]; the remaining studies used self-reported data and a 'traditional' analytical approach (i.e., did not take the co-dependency between physical activity types or intensities into account). Furthermore, all studies used general population samples, except three studies that used working populations [13,48,52]. Therefore, based on studies that have investigated how physical activity during both leisure and work are associated with SBP, the association between occupational physical activity and SBP is inconclusive.

Our results indicated a 1.7 (95% CI: -0.8, 4.2) mm Hg higher SBP given 30 minutes less sedentary behaviour and 30 minutes more walking during work, and an 0.7 (95% CI: -2.6, 1.2) mm Hg lower SBP given the same time reallocation during leisure. Furthermore, 30 minutes less walking and 30 minutes more sedentary behaviour during work suggested a 6.7 (95% CI: -16.2, 2.9) mm Hg lower SBP. This difference is 11 times larger than that of the opposite reallocation during leisure (i.e., 30 min less sedentary behaviour and 30 min more walking: -0.7, 95% CI: -2.6, 1.2 mm Hg), and could be expected to reduce the risk of CVD-specific mortality by over 20% based on the known linear relationship between SBP and CVD [54,55]. Since even small changes in the population mean SBP can have substantial impact on CVD risk (i.e., affecting the prevalence of hypertension) [54–56], these findings could, potentially, have important implications in population-based prevention of CVD [44].

**Waist circumference.**   During both domains, our results indicated less sedentary behaviour and more walking compared to the reference composition to be associated with a *larger* WC (Fig 3, Table 3). This finding may, potentially, be attributed to differences in occupation, socioeconomic status, and health, since low socioeconomic status is known to be associated with poor health [21], including overweight and dyslipidaemia [22]. That is, individuals with lower socioeconomic status who, in general, have poorer health are more likely to have occupations that involve little sedentary behaviour and high physical activity [18], such as long durations of walking. Further, we emphasise that the association between physical activity and overweight is bidirectional, and that other factors not considered in our analyses (e.g., diet) are influencing a person's WC. Importantly, these findings highlight that our estimates represent measures of associations, and not causal effects [57]. Furthermore, we found less sedentary behaviour and more HIPA during leisure to be associated with a smaller WC (e.g., 10 min less sedentary and 10 min more HIPA: -1.35, 95% CI: -1.90, -0.80 cm; Fig 3, Table 4). The estimates during work followed the same pattern but were small and the CIs included zero. This is in line with existing evidence from observational and intervention studies [58–61]. The current findings also support that domain-specific characteristics of physical activity do not affect risk factors for which diet is most important [62–64].

In previous studies based on total or leisure time physical activity, less sedentary behaviour and more physical activity, in particular HIPA, is reported to be associated with lower WC [58]. The results for WC in the present study are in agreement with this (i.e., given the reallocation of time from sedentary behaviour to HIPA in both domains). However, to our knowledge, few studies have investigated how both leisure time and occupational physical activity are associated with WC [47,51–53,65–67]. Only two of these studies used accelerometer-data [66,67], and one used iso-temporal substitution modelling [67]; the remaining studies used self-reported data and 'traditional' analyses. None of these studies found contrasting associations between leisure and work, although some only found associations during one of the two domains [47,51,66]. In the current study, the reallocation of time from sedentary behaviour to walking during both domains seemed to be associated with a larger WC, which is incongruent with one previous study that did not find an association between less sedentary behaviour and more walking [67]. On the other hand, the results in our study indicated an association between less sedentary time and more HIPA and a smaller WC, which is in line with two

previous studies [51,67]. Finally, in five studies that focussed on sedentary behaviour, the direction of the reported associations is varied, but the findings do not suggest that physical activity during leisure and work have contrasting associations with WC [47,52,53,65,66].

From a population-based prevention-perspective, even small shifts in the population mean of WC, such as the 1.4 cm smaller WC given 10 minutes less sedentary behaviour and 10 more minutes of HIPA during leisure, can have implications for public health, since it may decrease the prevalence of individuals at increased risk for CVD due to a high WC.

**Low-density lipoprotein cholesterol.** For LDL-C, during both domains the results indicated that less sedentary behaviour and more walking was associated with a higher LDL-C (Fig 4, Table 3). Similar to WC, and as previously discussed, one potential explanation to these findings may be confounding by socioeconomic status and occupation, which are linked to poor health [18,21,22]. Furthermore, during leisure and work, the results indicated that less sedentary behaviour and more HIPA was associated with a lower LDL-C (e.g., 10 minutes during leisure: -0.07, 95% CI: -0.12, -0.02 mmol/L; Fig 4, Table 4). This is in line with clinical guidelines, where leisure time physical activity is regarded to have a smaller effect on LDL-C (i.e., <5%) compared to, for example, high-density lipoprotein cholesterol (HDL-C) (i.e., >10%) [63]. These findings also support that differences in the characteristics of physical activity during leisure and work do not affect LDL-C differently. This is likely because LDL-C is mainly influenced by total energy expenditure, and not by type of physical activity, posture, or pattern of accumulation over time [62,63,68].

The results for LDL-C in the current study indicated different associations between the two reallocations but did not differ between domains (Fig 4, Tables 3 and 4). During both leisure and work, the reallocation of time from sedentary behaviour to walking suggested an association with a higher LDL-C. This is in agreement with one study only investigating occupational physical activity [49], but in disagreement with other studies that have investigated how sedentary behaviour or physical activity during both leisure and work is associated with LDL-C [51,52,65]. Furthermore, during both domains, less sedentary behaviour and more HIPA seemed to be associated with a lower LDL-C. This disagrees with findings from three studies [51,52,65], where similar associations were reported for sedentary behaviour during leisure but not for work (except for the study by Honda et al. [52] where indications of opposite associations during leisure and work are reported). All mentioned studies used self-reported data and 'traditional' analyses. Hence, given the results of our study and previous literature, the association between physical activity during leisure and work, and LDL-C is unclear.

On a population-level, a 1 mmol/L lower non-HDL-C (i.e., total cholesterol minus HDL-C) has been reported to lower IHD-mortality by 30% [69]. This translates to 0.3% lower IHD-mortality for every 0.01 mmol/L lower LDL-C. Therefore, even small improvements in LDL-C on a population-level like those observed in the current study, could, in combination with improvements in other modifiable risk factors (e.g., poor diet, high SBP, obesity, smoking, high alcohol consumption, and others), likely contribute to the prevention of incident IHD [70,71]. However, the potentially detrimental association between less sedentary behaviour and more HIPA during work and SBP should be kept in mind.

**Sensitivity analyses.** The results of the sensitivity analysis where those taking antihypertensives, diuretics, or cholesterol lowering drugs were included did not differ substantially from the main analysis (Table A-C in S3 File). However, the second sensitivity analysis indicated that the association between sedentary behaviour, walking, and HIPA during work and leisure, and SBP, WC, and LDL-C among those reporting the use of antihypertensives, diuretics, or cholesterol lowering drugs differed from those not taking these medications (Table D-F in S3 File). For example, the estimated differences in SBP for the sedentary behaviour and walk-reallocations were markedly larger during both domains. On the other hand, a pattern

opposite to the one found in the main analysis was observed for the sedentary behaviour and HIPA-reallocations. We emphasise that there were differences in the geometric mean (i.e., the starting points for the time reallocations) of the physical activity types between those taking and not taking antihypertensives, diuretics, or cholesterol lowering drugs. Specifically, those taking antihypertensives, diuretics, or cholesterol lowering drugs were on average more sedentary and less active during leisure but less sedentary and more active during work compared to those not taking these medications. This should be kept in mind when interpreting these results. Also, the lower number of individuals (n = 146) results in less precision of the estimates.

## Methodological considerations

Firstly, we emphasise that our estimates are based on cross-sectional data and should be interpreted as measures of association and not causal effects [57]. We also acknowledge the risk for reversed causality, in particular for WC because the relationship between physical activity and adiposity measurements appears to be bi-directional [72]. We also emphasise that the findings should be interpreted from a primary prevention perspective, since study participants reporting the use of antihypertensives, diuretics, and cholesterol lowering medicine were excluded because the use of these medications could modify the investigated relationships.

We know from a previous study, that those who accepted to wear accelerometers in the fifth examination of the CCHS and those who fulfilled our accelerometer data inclusion criteria are different than their counterparts on several characteristics [30]. In addition, those who fulfilled the inclusion criteria of this study differed from those not fulfilling the criteria. As in all epidemiological studies including working populations, a healthy worker effect may be present in the current study [73]. We acknowledge the apparent selection bias, but emphasise that representativeness is not an aim in itself [74,75] when estimating measures of association or investigating physiological mechanisms (i.e., where normal biological variation without the influence of 'external' factors, such as medication, is important) [74,75]. Furthermore, the results of the sensitivity analyses indicated that the exclusion of individuals taking antihypertensives, diuretics, or cholesterol lowering drugs did not influence the overall results. However, they indicated that the association between physical activity and sedentary behaviour during leisure and work, and risk factors for CVD may be different among individuals with pre-existing CVD.

Given the reported high sensitivity and specificity [32,33], we consider the validity and precision of Acti4's physical activity classification to be high. However, some details are important to emphasise when interpreting the results. Firstly, the measurements do not capture the load in specific tasks such as heavy lifting, pushing, pulling, or awkward body positions (does not include measurements of the weight of materials, people, or tools handled), which are known to impose high physical demands, and therefore, could be important [13,14]. Secondly, common to all accelerometer-based measurements of physical activity, the measurements do not include the relative intensity of the physical activity. Thirdly, we do not know whether the measurement period accurately reflects the study participants' typical physical activity level. Finally, we do not have data on job title, and on past or cumulative job exposure. These limitations imply a risk for misclassification of the exposure which, potentially, could lead to an underestimation of the health effects.

With regards to our outcomes, the risk that some SBP measurements were affected by white coat hypertension or masked hypertension should be acknowledged. This limitation could be overcome in future studies by the use of ambulatory blood pressure, which also seems to be a stronger predictor of CVD [76]. Furthermore, the magnitude of measurement error in

WC has been reported to be highly varying [77], which should be acknowledged. We used WC rather than BMI or waist-hip ratio since it has been suggested to be a stronger predictor for CVD [36]. Finally, our LDL-C measurements were based on non-fasting blood samples. Importantly, since habitual meals do not affect LDL-C to a significant degree [71,78], we do not believe this to have affected the precision of the LDL-C measurements. We chose LDL-C as a clinically relevant biomarker of dyslipidaemia due to its strong association with CVD risk and central role in the management of CVD (e.g., risk assessment and treatment target). Furthermore, the literature regarding the association between physical activity and LDL-C is inconclusive, and therefore, we believe our study can supplement existing knowledge.

It should be emphasised that the geometric mean time spent in the physical activity types was used as the reference composition in our time reallocations. Although in line with previous studies [26,27], one limitation with this approach is that the estimated outcome may be less accurate for study participants with a more extreme composition compared to the estimates of those whose composition lies closer to the reference. This is reflected in the wider confidence intervals seen for the time reallocations furthest away from the reference composition. Additionally, since the time spent in HIPA in general, and, in particular, during work was quite low in our study population, we could only investigate small time reallocations between sedentary behaviour and HIPA.

In general, the estimates were small, and the CIs were wide, in particular for the work-specific time reallocations. This is likely a consequence of the size of our study population, and the relatively small number of participants with a long duration of HIPA during work, which results in a large variation. A larger study population would likely result in less variation and thereby improved precision of the estimates, which could increase the confidence when interpreting the results.

## Perspectives

In general, all physical activity is considered to be health beneficial compared to sedentary behaviour. This is, for example, reflected in current physical activity recommendations [16,79], and the results of the current study support the importance of an active leisure for good health. However, as previous studies and our results indicate [8,12,80], public health messages such as 'sit less and move more', may not be well suited for population groups that are highly physically active during work. On the one hand, more leisure time physical activity may lead to increased fitness and workability (i.e., both physical and mental capacity), which could decrease the relative workload and thereby the risk of CVD and other non-communicable diseases. On the other hand, more leisure time physical activity may lead to cardiovascular overload and a vicious cycle of decreasing fitness over time; a scenario in which rest and restitution should be recommended. Currently, for several health outcomes it is still unclear how individuals with high occupational physical activity should best compensate during leisure. One potential alternative is workplace-based initiatives, such as aerobic exercise during work hours. Although such interventions may have unintended negative health effects such as increased SBP [81], they can improve cardiorespiratory fitness, workability, and health [81–83]. It is, therefore, highly important to take the potentially contrasting health effects of leisure time- and occupational physical activity into account in physical activity recommendations for adults.

This study exemplifies how a 24-hour approach that integrates different domains can improve our knowledge about how physical activity and health outcomes are associated, and the results highlight the importance of considering physical activity during both leisure and work. The results also reflect the fact that durations of physical activity types are co-dependent,

and that the association between a physical activity type and health outcome depends on how the day is composed and on what activity type an increase in one activity displaces. However, there is a need for studies with larger study samples and prospective data that further investigate the health effects of walking and other physical activity types during both leisure and work. Combining device-based measurements with data on previous job titles, job exposure matrices, routinely collected administrative data (e.g., periods of sick leave periods, retirement), or questionnaire data to improve the exposure assessment and minimise misclassification could be a fruitful avenue for future studies. There is also a need to better understand how existing knowledge can be implemented to increase physical activity levels in the population, and what to recommend to population groups with high occupational physical activity levels with regards to their leisure time physical activity.

## Conclusions

Less sedentary behaviour and more walking or HIPA seems to be associated with a lower SBP during leisure, but, during work, it seems to be associated with a higher SBP. In contrast, no consistent differences between domains were observed for WC and LDL-C. These findings highlight the importance of considering the physical activity health paradox, at least for some risk factors for CVD. The adverse health effects associated with occupational physical activity should inform physical activity recommendations.

## Supporting information

**S1 Table. Overview of questions and responses.**
(PDF)

**S2 Table. Overview of derived variables.**
(PDF)

**S3 Table. Variation matrix of parts in physical activity composition.**
(PDF)

**S4 Table. Comparison of characteristics of non-eligible and eligible participants.**
(PDF)

**S1 File. Linear regression models.**
(PDF)

**S2 File. Time reallocations.**
(PDF)

**S3 File. Sensitivity analyses.**
(PDF)

## Acknowledgments

PS established and designed the CCHS. PS and AH developed the initial idea and designed and funded the accelerometer measurements in the fifth round of the CCHS. MSJ, KS, AH and MK contributed to the conception and design of the present study. MSJ led the work with the processing of the raw accelerometer data, performed the analyses, the initial data interpretation, and formulated and developed the manuscript. AH, JLM, EP, PS, MK, and KS contributed with critical revising during the development of the manuscript. All authors have discussed the results and have given approval to the publishing of the final version of the

manuscript. We acknowledge the research personnel of the CCHS for their work with the data collection, research personnel at the National Research Centre for the Working Environment for their contribution in the processing and analyses of the accelerometer data, and all individuals in the fifth examination of the CCHS for their participation.

## Author Contributions

**Conceptualization:** Melker S. Johansson, Andreas Holtermann, Mette Korshøj, Karen Søgaard.

**Data curation:** Melker S. Johansson, Jacob L. Marott.

**Formal analysis:** Melker S. Johansson.

**Funding acquisition:** Andreas Holtermann, Mette Korshøj.

**Investigation:** Melker S. Johansson.

**Methodology:** Melker S. Johansson, Andreas Holtermann, Mette Korshøj, Karen Søgaard.

**Project administration:** Melker S. Johansson.

**Resources:** Andreas Holtermann.

**Software:** Andreas Holtermann.

**Supervision:** Andreas Holtermann, Mette Korshøj, Karen Søgaard.

**Validation:** Andreas Holtermann, Mette Korshøj, Karen Søgaard.

**Visualization:** Melker S. Johansson.

**Writing – original draft:** Melker S. Johansson, Karen Søgaard.

**Writing – review & editing:** Melker S. Johansson, Andreas Holtermann, Jacob L. Marott, Eva Prescott, Peter Schnohr, Mette Korshøj, Karen Søgaard.

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
