## [Decision Letter · Decision Letter 0]

12 Feb 2021

PONE-D-21-00560

The physical activity health paradox and risk factors for cardiovascular disease: a cross-sectional compositional data analysis in the Copenhagen City Heart Study

PLOS ONE

Dear Dr. Melker Staffan Johansson,

Thank you for submitting your manuscript to PLOS ONE. After careful consideration, we feel that it has merit but does not fully meet PLOS ONE’s publication criteria as it currently stands. Therefore, we invite you to submit a revised version of the manuscript that addresses the points raised during the review process.

Mainly discussion and interpretation of results and conclussions need clarifications and crrections. Some aspects of methodology should be also clarified and/or completed.

We look forward to receiving your revised manuscript.

Kind regards,

Pedro Tauler, Ph.D.

Academic Editor

PLOS ONE

Journal Requirements:

2)  We note that you have indicated that data from this study are available upon request. PLOS only allows data to be available upon request if there are legal or ethical restrictions on sharing data publicly. For information on unacceptable data access restrictions, please see http://journals.plos.org/plosone/s/data-availability#loc-unacceptable-data-access-restrictions.

3) We note that you have included the phrase “data not shown” in your manuscript. Unfortunately, this does not meet our data sharing requirements. PLOS does not permit references to inaccessible data. We require that authors provide all relevant data within the paper, Supporting Information files, or in an acceptable, public repository. Please add a citation to support this phrase or upload the data that corresponds with these findings to a stable repository (such as Figshare or Dryad) and provide and URLs, DOIs, or accession numbers that may be used to access these data. Or, if the data are not a core part of the research being presented in your study, we ask that you remove the phrase that refers to these data.

4) Please include captions for your Supporting Information files at the end of your manuscript, and update any in-text citations to match accordingly. Please see our Supporting Information guidelines for more information: http://journals.plos.org/plosone/s/supporting-information.

5)  Thank you for submitting the above manuscript to PLOS ONE. During our internal evaluation of the manuscript, we found significant text overlap between your submission and the following previously published works, some of which you are an author.

- https://www.researchsquare.com/article/rs-10805/v3

- https://kclpure.kcl.ac.uk/ws/files/115120458/accepted_plos_version.pdf

Please revise the manuscript to rephrase the duplicated text, cite your sources, and provide details as to how the current manuscript advances on previous work. Please note that further consideration is dependent on the submission of a manuscript that addresses these concerns about the overlap in text with published work.

Reviewers' comments:

Reviewer's Responses to Questions

**Comments to the Author**

1. Is the manuscript technically sound, and do the data support the conclusions?

Reviewer #1: Yes

Reviewer #2: Yes

2. Has the statistical analysis been performed appropriately and rigorously? 

Reviewer #1: Yes

Reviewer #2: Yes

3. Have the authors made all data underlying the findings in their manuscript fully available?

Reviewer #1: Yes

Reviewer #2: Yes

4. Is the manuscript presented in an intelligible fashion and written in standard English?

Reviewer #1: Yes

Reviewer #2: Yes

5. Review Comments to the Author

Reviewer #1: 1. The study presents the results of original research. THIS IS ORIGINAL RESEARCH

2. Results reported have not been published elsewhere. RESULTS ARE NOT PUBLISHED BEFORE

3. Experiments, statistics, and other analyses are performed to a high technical standard and are described in sufficient detail. ALL THESE ARE WELL EXPLAINED

4. Conclusions are presented in an appropriate fashion and are supported by the data. ARE STILL INSUFICIENTS, CAN BE IMPROVED

5. The article is presented in an intelligible fashion and is written in standard English. WELL DONE

6. The research meets all applicable standards for the ethics of experimentation and research integrity. YES

7. The article adheres to appropriate reporting guidelines and community standards for data availability. YES

FOR MORE DETAILS, PLEASE SEE THE ATTACHED DOCUMENT

Reviewer #2: See attached file

6. PLOS authors have the option to publish the peer review history of their article (what does this mean?). If published, this will include your full peer review and any attached files.

Reviewer #1: No

Reviewer #2: No

---

## [Author Response · Author response to Decision Letter 0]

16 Jan 2022

Response to reviewers’ comments

PONE-D-21-00560

The physical activity health paradox and risk factors for cardiovascular disease: a cross-sectional compositional data analysis in the Copenhagen City Heart Study

Dear Dr. Pedro Tauler,

Thanks for letting us revise our manuscript. We would like to acknowledge the reviewers for taking their time to assess our manuscript and providing valuable feedback. A point-by-point response can be found below. Changes made in the manuscript have been highlighted using the track-changes function in Word. 

Reviewer 1

To be accepted for publication in PLOS ONE, research articles must satisfy the following criteria:

1. The study presents the results of original research. THIS IS ORIGINAL RESEARCH

2. Results reported have not been published elsewhere. RESULTS ARE NOT PUBLISHED BEFORE

3. Experiments, statistics, and other analyses are performed to a high technical standard and are described in sufficient detail. ALL THESE ARE WELL EXPLAINED 

4. Conclusions are presented in an appropriate fashion and are supported by the data. ARE STILL INSUFICIENTS, CAN BE IMPROVED 

5. The article is presented in an intelligible fashion and is written in standard English. WELL DONE

6. The research meets all applicable standards for the ethics of experimentation and research integrity. YES 

7. The article adheres to appropriate reporting guidelines and community standards for data availability. YES 

Background 

1. The purpose of the study needs to be fixed (talk about objectives, believe there is one)

Supplementary information received after contacting the editor: “Regarding the comment from the reviewer, the aim of the study should be clearly stated taking into account previous background reported and the lack of studies in the field. No reason has been provided to include in the aim that "The measure of association was quantified by reallocating time between 1) sedentary behaviour and walking, and 2) sedentary behaviour and HIPA, during leisure and work.", which, in turn, seems information more adequate about the study desing. Furthermore, it is not clear from the aim whether, for example, sitting time and walking time would be considered together in the analysis as dependent variables, or each one would be considered in different analysis.“ 

Response: We acknowledge that the use of time reallocations relates more to the methods section and have deleted it from the objectives. It is described in the statistical analysis-section on p. 12-13.

In the present study, the durations of the specific physical behaviours are the independent variables (i.e., the explanatory variables) and the three risk factors for CVD: systolic blood pressure, waist circumference, and low-density lipoprotein cholesterol are the dependent variables (i.e., the outcomes). Each of these outcomes are considered in the present study population in three separate analyses. 

Action: We have omitted the following sentence: “The measure of association was quantified by reallocating time between 1) sedentary behaviour and walking, and 2) sedentary behaviour and HIPA, during leisure and work.” from the introduction.

Since we clearly distinguish the physical behaviour composition (i.e., the exposure or explanatory variables) from the outcomes in the statistical analysis section (e.g., p. 12, line: 265-267 and 269-271), no action has been taken in relation to the question about the dependent variables of the study.

Results

2. First paragraph> some data within the text are repeated in Table1, and again for table 2

Response: We acknowledge that some data in Table 1 and Table 2 have been repeated in the first and second paragraph, respectively, of the results section. 

Action: The text in the first paragraph has been shortened to: “We have illustrated the cohort formation in Figure 1 and presented characteristics of the study population in Table 1. The median number of valid days was 6 and the study participants wore the accelerometers for a median time of 23.8 h/day. Furthermore, the median number of workdays was 4 and 94% had >1 workday. The median worktime was 7.6 h/day. There were 58% women, and the median age was 48.6 years. The median SBP, WC, and LDL-C was 128 mm Hg, 83 cm, and 3.0 mmol/L, respectively.” (p. 14, line: 311-316).

Similarly, the second paragraph has been shortened to “The geometric mean of each part of the physical activity composition is presented in Table 2, stratified by leisure and work.” (p. 16, line: 319-320).

3. Line 336> how was the judgement of the models/residuals? That may be important to explain a little bit more

Response: We examined the distribution of the residuals using quantile-quantile (Q-Q) plots of standardized residuals from the regression models. Any deviations from the projected solid line indicates non-normal distributions (in particular shapes similar to a hammock). In an ideal world, the residuals follow the line; however, small deviations are not uncommon in real-world data but may not compose a problem. We acknowledge that this is a subjective process and have therefore included the Q-Q plots in Figure A-C in the Supporting information File S1 (as part of the original submission). However, “judged” may be poor wording and we have therefore paraphrased the sentence. 

Action: It now reads: “For all three models, the residuals were not perfectly normally distributed, but the deviations were considered too small to substantially affect the model fit.” (p. 17, line: 333-335).

4. In my opinion, tables 3 and 4 are huge, and that makes a little bit harder to understand what is there.

Response: We acknowledge that Table 3-4 are large as they both contain all results from the time reallocations across the three outcomes. However, in order to facilitate comparisons across outcomes and domains, we prefer to keep this condensed presentation. 

Action: None.

Discussion 

5. Major concern> Results indicated that less SB and more walking to be associated with a larger WC. This is a major finding contradicting what is known. The explanation of this finding needs to be addressed deeper making a better case. What is already explained is confusing. Occupations that involve less SB and long walking (more PA) may impact WC in a positive way independently of socioeconomic status. I would like to see more on this to be more convincing. Low income is associated with poor health, but I am not sure that low-income people who work in environments that include high PA would have larger WC. In my opinion it is important to identify other variables or circumstances that may have some associations with the previous finding to explain, convince, and make a better case. 

6. LDL-C> same as above. These 2 results need to be deeper explained 

7. Compensation and nutrition may have something here in both variables

Response comment 5-7: 

Firstly, this study is based on cross-sectional data and does, therefore, not show causal effects but associations. Secondly, the time reallocations are conducted to quantify the investigated associations since the beta-coefficients of the ilr-coordinates cannot be interpreted directly in the same way as a linear regression model fitted with non-compositional variables (due to the ilr-transformation). In essence, the time reallocations show the predicted difference in the mean outcome (i.e., at a population level) given theoretical changes in the mean physical behaviour composition. Therefore, this does not reflect changes in the outcomes on an individual level but reflects that, in this cohort, individuals who sit less and walk more during work compared to those with an average physical behaviour composition have a larger WC and higher LDL-C.

Re “Occupations that involve less SB and long walking (more PA) may impact WC in a positive way independently of socioeconomic status.”: We believe there has been a misunderstanding. This is not what we mean. Occupations that involve low levels of sedentary behaviour and longer duration of walking are most often held by individuals with lower socioeconomic status, which is associated with poorer health such as obesity and dyslipidaemia. That is, our results could be confounded by socioeconomic status, despite the fact that we have tried to adjust for this by including level of education in the regression models.

Action comment 5-7: We have paraphrased the discussion related to WC to soften the description of the potential explanation to the findings. It now reads: “During both domains, our results indicated less sedentary behaviour and more walking compared to the reference composition to be associated with a larger WC (Figure 3, Table 3). This finding may, potentially, be attributed to differences in occupation, socioeconomic status, and health, since low socioeconomic status is known to be associated with poor health (22), including overweight and dyslipidaemia (23). That is, individuals with lower socioeconomic status who, in general, have poorer health are more likely to have occupations that involve little sedentary behaviour and high physical activity (18), such as long durations of walking. Further, we emphasise that the association between physical activity and overweight is bidirectional, and that other factors not considered in our analyses (e.g., diet) are influencing a person’s WC. Importantly, these findings highlight that our estimates represent measures of associations, and not causal effects (58).” (p. 26, line: 501-511). 

Similarly, we have elaborated the discussion related to LDL-C. It now reads: “For LDL-C, during both domains the results indicated that less sedentary behaviour and more walking was associated with a higher LDL-C (Figure 4, Table 3). Similar to WC, and as previously discussed, one potential explanation to these findings may be confounding by socioeconomic status and occupation, which are linked to poor health (18, 22, 23).” (p. 28, line: 544-547).

Methodological considerations

8. LDL was chosen due to the relationship with CVD, however, HDL could be more important due to the protective effect, and also because PA seems to have incremental effects on HDL levels but less effects on LDL. In this case, it may be important to explain why HDL was not used as variable.

Response: As we mention in the discussion, we chose LDL-C because it is most clinically relevant as a risk factor for CVD and plays a more central role in the management of CVD (e.g., risk calculation) than HDL. In addition, only few studies have investigated LDL and the association to device-based measurements of physical behaviours.

We agree that several other relevant biomarkers could have been chosen but have limited our focus to the current three risk factors, to restrict the number of analyses.

Action: We have clarified the choice of LDL. It now reads: “We chose LDL-C as a clinically relevant biomarker of dyslipidaemia due to its strong association with CVD risk and central role in the management of CVD (e.g., risk assessment and treatment target). Furthermore, the literature regarding the association between physical behaviours and LDL-C is inconclusive, and therefore, we believe our study can supplement existing knowledge.” (p. 32, line: 647-651).

Perspectives

9. From line 572 to line 584, if compensation for PA is included, it would help to understand people’s behavior and some results from the study.

Response: We acknowledge that physical behaviours in the two domains could to some extent compensate for each other. This is now incorporated in the paragraph. 

Action: It now reads: However, as previous studies and our results indicate (8, 12, 80), public health messages such as ‘sit less and move more’, may not be well suited for population groups that are highly physically active during work. On the one hand, more leisure time physical activity may lead to increased fitness and workability (i.e., both physical and mental capacity), which could decrease the relative workload and thereby the risk of CVD and other non-communicable diseases. On the other hand, more leisure time physical activity may lead to cardiovascular overload and a vicious cycle of decreasing fitness over time; a scenario in which rest and restitution should be recommended. Currently, for several health outcomes it is still unclear how individuals with high occupational physical activity should best compensate during leisure. One potential alternative is workplace-based initiatives, such as aerobic exercise during work hours. Although such interventions may have unintended negative health effects such as increased SBP (81), they can improve cardiorespiratory fitness, workability, and health (81-83). It is, therefore, highly important to take the potentially contrasting health effects of leisure time- and occupational physical activity into account in physical activity recommendations for adults. (p. 36, line: 673-687).

Conclusions

10. I would like to see the take home message here and not the already known results from the study. What is the impact of the study, what it apports to the knowledge, why is important to consider SB, walking, and HIPA during leisure and work.

Response: We have paraphrased the conclusion towards a clearer take home message. 

Action: The conclusion now reads: “Less sedentary behaviour and more walking or HIPA seems to be associated with a lower SBP during leisure, but, during work, it seems to be associated with a higher SBP. In contrast, no consistent differences between domains were observed for WC and LDL-C. These findings highlight the importance of considering the physical activity health paradox, at least for some risk factors for CVD. The adverse health effects associated with occupational physical activity should inform physical activity recommendations.” (p. 34, line: 706-711).

Correspondingly, we have paraphrased the conclusion in the abstract. It now reads: “During leisure, less sedentary behaviour and more walking or HIPA seems to be associated with a lower SBP, but, during work, it seems to be associated with a higher SBP. No consistent differences between domains were observed for WC and LDL-C. These findings highlight the importance of considering the physical activity health paradox, at least for some risk factors for CVD.” (p. 2-3, line: 45-49).

 

Reviewer 2

RE: MS PONE-D-21-00560 The physical activity health paradox and risk factors for cardiovascular disease: a cross-sectional compositional data analysis in the Copenhagen City Heart Study

Review, January 28, 2021

General Comments:

1) This study of differential associations between domain-specific leisure time and occupational physical activity with three common cardiovascular disease risk factors addresses an important occupational and public health issue: identification of potentially modifiable underlying mechanisms of the emerging physical activity health paradox using innovative physical activity exposure assessment (wearable sensors employing accelerometry), appropriate differentiation of work and leisure, and innovative analytic approaches estimating effects using compositional data analysis. This is a seminal contribution to the evolving literature regarding the PA health paradox and deserves publication in a high-quality journal.

2) Additional major strengths of this manuscript include an obvious command over the most relevant literature in this field and appropriate citations throughout. Only few modifications are recommended (see details below). The paper is very well written, concise, and in virtually flawless English language. Provision of additional details in supplemental files are also noted as a positive feature.

3) The major acknowledged limitations include the cross-sectional design and potential sample selection bias. There are several other limitations that deserve to be discussed: conservative biases leading to underestimation of health effects due to exposure misclassification, healthy worker effects, and exclusion of eligible participants with pre-existing health conditions (such as IHD or hypertension) that are known to modify the health effects of both leisure and occupational physical activity.

4) Since data on pre-existing conditions appear to be available (these data have been used to exclude up to 2/3 of study participants) respective subgroup analyses appear to be possible within the available data set. This reviewer strongly recommends to provide additional sensitivity analyses using a larger sample that does not exclude these eligible participants and to also supplement stratified analyses in respective subsamples based on common pre-existing conditions such as IHD and hypertension that have been shown to be effect modifiers in earlier research.

5) Use consistent terminology: The title refers correctly to the technical term “physical activity health paradox,” however abstract and text often instead use “physical behavior.” I would recommend to stay consistent with the title throughout the manuscript and also with the decades-old research literature and consistently use the term “physical activity.”

In cardiovascular research the term “behaviour” is considered to point to activities over which the individual has control and thus is responsible for and able to “change behavior.” While this label has been applied in CVD research to smoking, drinking, and also leisure time physical activity, it has the tendency to distract from the social determinants of even the most private individual behaviors, it but it definitely not a good choice to characterize occupational physical activity where the activity is determined by the physical environment, explicit employer direction (e.g. the mandate to stand when serving customers in a bank even if the job could be performed sitting), or inherent in the job task itself and thus to a large extent out of the control of the individual worker. These may be subtle distinctions, however, given the long history of occupational medicine where inherently unsafe working conditions were ascribed to random acts of nature (“accidents”) or individual personal worker traits or behaviors (“accident-prone worker”, “unsafe behavior”) in order to abdicate employer responsibility and liability, shifting the more neutral term (with regard to agency/control) of “activity” to “behavior” may be perceived as implicitly “blaming the victim” – a historical legacy burden in occupational medicine and even public health that had many disastrous consequences for legions of workers, their families, and their communities in the past centuries and even today.

Response to general comments 1-5: We appreciate the kind words and thank reviewer 2 for the thorough review, which we believe have helped us improve the manuscript. Since there is an overlap between the general comments and the detailed comments, we have focused on providing specific responses and actions to the 38 detailed comments below. 

Briefly, regarding 3), we agree that there are several additional limitations that can be discussed. Conservative bias has been addressed in relation to detailed comment 29 (discussion of potential exposure misclassification). Healthy worker effect has been addressed; see action below. The exclusion of participants with pre-existing health conditions have been addressed in relation to detailed comment 6 and 7. 

Regarding 4), we used self-reported use of medication as a proxy for pre-existing health conditions. The suggested sensitivity analyses have been conducted and added to the Supplementary files.

Regarding 5), the “physical activity health paradox” refers to different health effects from leisure time physical activity and occupational physical activity. The use of terminology has been addressed in relation to detailed comment 1. 

Action to general comments 1-5: We have added a sentence acknowledging the potential of a healthy worker effect. It reads: “As in all epidemiological studies including working populations, a healthy worker effect may be present in the current study (73).” (p. 30, line: 615-615). 

Specific actions in response to general comment 3, 4, and 5 are described in the detailed comments below.

Detailed Comments:

Abstract:

1) Line 32: Use consistent terminology: The title refers correctly to the technical term “physical activity health paradox,” however abstract and text often instead use “physical behavior.” I would recommend to stay consistent with the title throughout the manuscript and also with the decades-old research literature and consistently use the term physical activity. 

Aside: Additional comment addressing the broader research context: It is important to note that in cardiovascular research the term “behaviour” is used to indicate activities over which the individual has control and thus is in general considered responsible for and assumed to have the ability to “change behavior.” In medical and epidemiological cardiovascular disease research this label has been applied consistently to smoking, drinking, and also leisure time physical activity, under the heading “health behaviors”. While this labeling has the unfortunate tendency to distract from the social determinants (that have been identified for even the most private individual behaviors such as suicide, see Durkheim’s 19th century seminal study of the same title), this is a widely accepted convention. However, to characterize OPA as a “behavior” is a problematic choice because the physical activity at work is mostly not a choice but instead determined by the physical and organizational work environment, explicit employer direction (e.g. the mandate to stand when serving customers if the job could be performed sitting), or inherent in the job task itself. Thus the intensity of OPA, its duration, and the work/rest/cycle is effectively out of the control of the individual worker, especially those who are performing high levels of OPA or repetitive tasks. Accordingly, the occupational health literature has been referred to PA at work as “occupational physical activity”, “physical workload”, “physical job demands” as more appropriate terms for most paid labor than terms that have connotations of individual choice or even a moral undertone like in “good or bad behavior.” 

These are subtle distinctions, however, OPA happens in social context where performing heavy work is associated with low status, low pay, excessive health and mortality risks. Behavioralism, “the advocacy or adherence to a behavioral approach to social phenomena” as defined in the dictionary combined with a tragic history of little attention by academia, and a long and shameful history of occupational medicine where unsafe working conditions, safety hazards inherent in a specific job design or work task have been attributed to random acts of nature (“accidents” instead of “work injury”) or individual personal worker traits (“accident-prone worker”), or behaviors (“unsafe behavior”, “worker negligence,” “human factor,”) or even a kind of mental illness (“accident neurosis” “Unfallneurose,” “pension neurosis,” “Rentenneurose”) if the victim of a work-injury demands compensation - these “behaviors” (?!) of medical professionals, academicians, scientists, and legal experts ignore the root causes of work-related injuries and illnesses and collude with regulators’ and/or employers’ attempts to deny their responsibilities for providing a safe work environment and their legal liability to compensate their injured workers for lost income, health, limbs, or life. Shifting the more neutral (with regard to agency/control) term of “activity” to “behavior” (with a connotation of worker choice and good/bad or health/unhealthy behavior) may thus be a subtle form of shifting the burden of work-related injury, illness, and disability or death and the legal mandate for primary prevention at the workplace onto the individual worker and thus implicitly “blaming the victim” – a historical legacy burden in occupational medicine and even public health that had many tragic consequences for legions of workers, their families, and their communities in the past centuries and even today.

Response: We agree that physical activity and stationary behaviours during work are for many individuals not a matter of choice but determined by work demands and organisation. We used the term physical behaviours as an umbrella term encompassing physical activity (i.e., any bodily movement produced by skeletal muscles that results in energy expenditure) and stationary behaviours (e.g., sedentary behaviour and standing; that is, behaviours that do not involve any movement). However, we acknowledge the possibility for misunderstandings and have therefore followed the advice and changed the terminology used.

Action: We have changed “physical behaviours” to “physical activity” or “physical activity and sedentary behaviour” throughout the entire manuscript (highlighted with tracked changes).

Background:

2) Line 71-74: The way references are inserted in this summary of the literature is confusing. The 2018 meta-analyses by Coenen et al. is cited as if it represents and individual study and for providing evidence for “beneficial health effects” and “no association with all-cause mortality” while this is a recent review that did not investigate health effects but only all-cause mortality (which is a different outcome) and actually concluded in the abstract: “Conclusions The results of this review indicate detrimental health consequences associated with high level occupational physical activity in men, even when adjusting for relevant factors (such as leisure time physical activity). These findings suggest that research and physical activity guidelines may differentiate between occupational and leisure time physical activity.” I would recommend to rewrite this summary, clearly differentiating between CVD risk factors, and CVD/IHD incidence, and cardiovascular and all-cause mortality, between reviews and individual studies, and between older reviews and newer reviews, because you are referring to “currently inconclusive” results.

Results regarding traditional CVD risk factors may be more inconclusive, results regarding all-cause mortality are more conclusive, at least for men. There is also a development in the literature: reviews of more recent studies and of higher methodological quality conclude that OPA effects on different outcomes are detrimental (Li 2013, Coenen 2018). 

Response: We acknowledge that the use of references in the summary of the literature could be confusing. We have followed the reviewers suggestions and rewritten it as a short and clear introduction based on recent reviews and recently published individual studies.

Action: The summary now reads: “Leisure time physical activity has well-established health benefits (1). For example, walking, cycling, and running, are considered to have favourable effects on risk factors for cardiovascular disease (CVD) and to lower the risk of mortality (2-7). However, emerging evidence indicate that occupational physical activity is associated with an increased risk of all-cause mortality, at least among men (8-10). Further, results from individual studies and literature reviews on the risk of ischemic heart disease (IHD) and major cardiovascular events from occupational physical activity are mixed (9-12). Similarly, the evidence regarding occupational physical activity and risk factors for CVD is currently inconclusive (10, 13, 14).” (p. 3, line: 55-63).

3) Line 81-82: This paragraph is clearly written, however, the last sentence starts with “therefore” but is not clearly stated why we should investigate “how,” by which mechanism, OPA affects health outcomes. (To understand it better and/or to identify additional points of interventions along the causal chain leading form OPA to health outcomes? Or to confirm biological pathways and plausibility? Or any other good reason you want to emphasize?)

Response: The “how” should have been an “if”.

Action: We have paraphrased the paragraph. It now reads: “… Therefore, it is important to investigate if occupational physical activity is associated with health outcomes in addition to other domains.” (p. 4, line: 70-71). 

4) line 90: “importantly” twice?

Response: We have paraphrased to improve readability. 

Action: The sentence now reads: “These differences in physical activity may be important for the effects on some risk factors (e.g., SBP), while not influencing risk factors that depend more on total energy expenditure (20), such as waist circumference (WC) or low-density lipoprotein cholesterol (LDL-C).” (p. 4, line: 79-81).

Methods:

5) line 179-190: I think this is a clear and important strength of this study: using two accelerometers and advanced algorithms that can differentiate between these different activities (with the possible exception of climbing stairs). What are the respective values of sensitivity and specificity for biking?

Response: The sensitivity and specificity for cycling have been found to be 99.9 and 100.0 during standardised conditions (1). 

Action: None.

6) line 209-214: Eligibility criteria: Exclusion of individuals using anti-hypertensives, diuretics, or cholesterol-lowering drugs is problematic for several reasons: (1) it excludes a large percent of the study population and thereby further limiting representativeness of an already highly selected sample. (2) Any exclusion based on these medications need to be justified for each outcome separately. Does it make sense to exclude anti-cholesterol meds when examining SBP outcome, or anti-cholesterol drugs when examining WC? In the former case, one may have not have excluded people with hypertension (if this was intended) but rather those who were not diagnosed or did not seek or get or adhere to treatment… etc. these choices are likely to introduce different selection biases that need to be addressed, if not here, at least in the discussion section (3) Most importantly, this will systematically exclude persons with some common pre-existing cardiovascular health conditions (treated hypertension and some other treated CVD) which have been shown to strongly interact with OPA in previous epidemiological research (see for example large differences in HRs in Table 4 in the Hall 2019 study you cite, and 4 other studies cited the method section of this paper justifying this approach). Did you collect data on persons with those conditions? The results section and Figure 1 seem to imply this. In fact, these exclusions lead to a loss of nearly 70% participants (1367 out of 2009 participants) due to a combination of these exclusion criteria and one unrelated factor (minimum wear time of sensors). Given these large numbers, it is important to breakdown the n for each exclusion criterion and add this in Figure 1 and/or text.

Response:

1) The exclusion criteria excluded 643 (32%) of the 2019 study participants with accelerometer data: 545 used antihypertensives (incl. diuretics), 297 used cholesterol lowering drugs, and 199 used both drugs. Importantly, not excluding those using antihypertensives or cholesterol lowering drugs would result in 152 study participants more (i.e., 804 in total). Furthermore, these exclusion criteria were chosen as an attempt to better isolate the potential association between the physical behaviour composition and the three risk factors among untreated, apparently healthy individuals. For transparency regarding selection bias, the differences in characteristics between included and excluded individuals are described in the results section (p. 16-17, line: 324-329) and presented in Table S4 in the Supplementary files.

2) Re “Does it make sense to exclude anti-cholesterol meds when examining SBP outcome, or anti-cholesterol drugs when examining WC?”: The main reason for applying the same inclusion criteria for all three outcomes (i.e., using one study population) was to avoid having three slightly different study populations, which potentially could have confused the reader. We would like to emphasise that there is a substantial overlap since 67% of those using cholesterol lowering drugs also used antihypertensives. 

3) We have data about self-reported conditions and self-reported use of medications on the excluded observations. The latter is used as a proxy for pre-existing cardiovascular disease. Also, we agree that it is important to show the number of individuals fulfilling each exclusion criteria.

Action: We have performed sensitivity analyses to investigate the influence of excluding individuals taking antihypertensives, diuretics, or cholesterol lowering drugs on the results. The results of these are briefly described in the results section (p. 22-23, line: 416-428) and presented in Table A-F in Supplementary file S3. See action to detailed comment 7 for further details. 

Finally, we have added the number of individuals excluded for each specific exclusion criteria in Figure 1.

7) ibid: On the other hand, given the large sample of persons with these conditions, possibly the majority of the original study population, I would strongly suggest to examine the potential of selection bias by performing respective sensitivity analyses stratifying by these conditions and also examine how overall results would change if you included these people in the main analysis with all participants who participated and who were otherwise eligible. This is not only important for assessing quantitatively the potential risk of selection bias but even more important because 21st century working populations in developed countries typically include a large percentage of workers with such conditions or medications and we need to understand if those workers experience different PA health effects in order to develop safe PA recommendations that not only differentiate between OPA and LTPA but also between healthy workers and the increasing number of workers with those conditions and in order to tailor any recommendations and interventions accordingly. 

Response: We see the reviewer’s point and have conducted the suggested sensitivity analyses. Specifically, we have conducted sensitivity analyses among a) study participants regardless of medication use (i.e., no exclusion of individuals taking antihypertensives, diuretics, or cholesterol lowering drugs), as well as b) among participants taking antihypertensives or cholesterol lowering drugs. Due to the low number of individuals and the large overlap (67%), we have not separated the analyses by the two types of drugs. The results are presented in Table A-F in Supplementary file S3.

Actions: In the methods section, we have added the following description of the sensitivity analyses: “To investigate the influence of excluding individuals taking antihypertensives, diuretics, or cholesterol lowering drugs, we conducted sensitivity analyses including all study participants regardless of medication use, and among those with the medications use, respectively.” (p. 14, line: 302-304).

In the results section, we have added the following: “Similar results were observed across the three outcomes when study participants taking antihypertensives, diuretics, or cholesterol lowering drugs were included in the analyses (Table A-C in File S3). When the analyses were limited to those taking these drugs (n=146), the estimated differences in SBP for time reallocations between sedentary behaviour and walking followed the same pattern but were larger than in the main analysis. However, the estimated differences in SBP given time reallocations between sedentary behaviour and HIPA followed an opposite pattern compared to the main analysis (Table D in File S3). Opposite patterns were also found for WC and LDL-C. Specifically, for WC in the sedentary-behaviour and walk-reallocations during leisure and the sedentary behaviour and HIPA-reallocations during work, and for LDL-C, in the sedentary-behaviour and walk-reallocations during both domains and in the sedentary behaviour and HIPA-reallocations during work (Table E and F in File S3).” (p. 22-23, line: 417-428).

In the discussion, we have added the following: “The results of the sensitivity analysis where those taking antihypertensives, diuretics, or cholesterol lowering drugs were included did not differ substantially from the main analysis (Table A-C in File S3). However, the second sensitivity analysis indicated that the association between sedentary behaviour, walking, and HIPA during work and leisure, and SBP, WC, and LDL-C among those reporting the use of antihypertensives, diuretics, or cholesterol lowering drugs differed from those not taking these medications (Table D-F in File S3). For example, the estimated differences in SBP for the sedentary behaviour and walk-reallocations were markedly larger during both domains. On the other hand, a pattern opposite to the one found in the main analysis was observed for the sedentary behaviour and HIPA-reallocations. We emphasise that there were differences in the geometric mean (i.e., the starting points for the time reallocations) of the physical activity types between those taking and not taking antihypertensives, diuretics, or cholesterol lowering drugs. Specifically, those taking antihypertensives, diuretics, or cholesterol lowering drugs were on average more sedentary and less active during leisure but less sedentary and more active during work compared those not taking these medications. This should be kept in mind when interpreting these results. Also, the lower number of individuals (n=146) results in less precision of the estimates.” (p. 29-30, line: 583-599).

In relation to the discussion of selection bias, we have added the following: “Furthermore, the results of the sensitivity analyses indicated that the exclusion of individuals taking antihypertensives, diuretics, or cholesterol lowering drugs did not influence the overall results. However, they indicated that the association between physical activity and sedentary behaviour during leisure and work, and risk factors for CVD may be different among individuals with pre-existing CVD.” (p. 31, line: 619-623).

8) Line 216-223: The descriptors for different PA composites used in the text do not correspond to the answer categories shown in Supplemental Table A1 for OPA and LTPA questions (e.g. I cannot find the item “climbing stairs” in that table). Since this is part of your key exposure variables, please provide a detailed account of all related questionnaire items and the specific re-coding or combinations you used.

Response: We emphasise that the analyses in this study are all based on device-based measurements of physical behaviours during leisure and work. The data presented in Table S1 defines questions and responses of the self-reported variables (i.e., collected with questionnaire, such as self-reported LTPA and OPA). However, these data were not used in the current analysis and should therefore not be included in the table. This information has survived from a previous iteration of the manuscript and should have been deleted. We apologise for the confusion.

Action: The information about the questions and responses regarding LTPA and OPA in Table S1 has been deleted.

9) line 226: Since BP is your key outcome measure, you should provide more detail about its measurement, specifically, if your protocols adhered to any of the standard guidelines for blood pressure measurements.

Response: As the study is the fifth examination of a large general population study, the protocol for the clinical tests followed procedures from earlier examinations to make valid comparisons possible. The test procedure follows, in general, the recommendations outlined in the 2020 International Society of Hypertension Global Hypertension Practice Guidelines (2).

Action: We have added some information to the description of the blood pressure measurements. It now reads: “Three blood pressure measurements were taken on participants’ non-dominant arm using an automatic blood pressure monitor (OMRON M3, OMRON Healthcare, Hoofddorp, Netherlands) after five minutes rest in a seated position. This test procedure has been used in previous examinations of the CCHS and is in line with the 2020 International Society of Hypertension Global Hypertension Practice Guidelines (32).” (p. 7, line: 145-150).

10) line 275: Not sure what you mean by “results for the daily physical behavior composition as a whole” – please show and explain data in your response to this review and/or as supporting information in the appendix

Response: We acknowledge that the sentence was confusing and have decided to omit it since it is not essential for the interpretation of the results of this study.

Action: The sentenced referred to has been omitted to avoid confusion. 

11) line 283-286: “One-to-one-reallocations” separately within work and within leisure seems to be very appropriate and actual crucial for your study aims and appropriate for future interventions since both domains are highly separated in terms of degree of self-determination and require different intervention strategies. This step has helped me to re-evaluate the promise of composition analysis in this field, in early formulations I saw, this was not emphasized or I missed it. Have your PA-intensity and domain-specific approaches used by others or have most other researchers reallocated PA across intensity levels and across domains? It may be worth highlighting your approach if it is innovative as such here (providing the rationale) and in the discussion (comparing with others and pointing to consequences), as this seem to me a major contribution of your paper. 

Response: We agree that separate time reallocations within work and leisure, respectively, are appropriate. To our knowledge, this is what most previous studies using compositional data analysis have done. Some previous studies have, however, conducted time reallocations involving more than two physical behaviours (i.e., “many-to-one reallocations”). 

Action: None.

12) I am pleased to see that this study defines “sedentary behavior” at work as sitting and does not include light standing/walking work because the common practice of many researchers combining sitting and “light standing” work into a common category “sedentary” is problematic because there is strong evidence that standing alone (with some walking but an mostly upright work posture) is a potentially strong risk factor for progression of atherosclerosis and CVD and mortality compared to sitting (even after adjustment for SES and other potential confounders (see Canadian study by Peter Smith, 2018 on AMI, and several Finnish studies by Krause et al. and Hall 2019 whom you cite). Combining sitting with standing causes a strong conservative misclassification bias by inflating the baseline risk in any reference group that contains standing work in addition to sitting work. Question: Could you assess the magnitude of this potential misclassification bias by a (not recommended) sensitivity analysis that uses “sitting or standing at work” (without lifting) as the reference group? That would be a nice extra contribution to the field and could be put into the appendix for a reference for other researchers. Your study, with objective measurements would be uniquely able to compare reference groups based on sitting alone or sitting and standing combined.

Response: We agree with the reviewer that standing may be a risk factor for CVD and that it should not be merged with sedentary behaviour. However, we feel that this question may deserve a more thorough investigation which lies beyond the scope of the present paper.

Action: None.

Results: 

13) line 307 ff and Table 1: The text and tables describe the distribution of study characteristics as medians and the first and third quartile. Is there any specific reason for this choice? Regardless, while these descriptors are not wrong and could still be provided in the appendix, this table should show the means the full range of all continuous variables to allow the reader to better compare this study population and its exposures and other covariates with other study populations in the literature that typically show means and full ranges.

Response: The specific reason for using medians and first and third quartiles (stated in the methods section, p. 11, line: 246-247), was skewed distributions of some of the continuous variables. We consider medians to be more appropriate measurements of the central tendency in these cases and have kept Table 1 as is. We do, however, acknowledge that mean with standard deviation are commonly used in summary statistics of study populations sometimes even in spite of non-normal distributions. 

Action: None.

14) line 345-49 Results for SBP: The description of results focuses on the patterns of point estimates of change in terms of the direction of change, which are the most important results. However, the rest of the description ignores the second most important data: effect sizes and instead implicitly focuses on statistical significance testing (using CIs as a substitute for p-values) while ignoring additional substantive quantitative information contained in both point estimates and CIs.

For example, reallocation from reference to more sedentary behavior shows only small increases in BP between +0.21 and +0.60 mmHg for LTPA but rather substantial decreases from -0.95 to -6.66 mmHG for OPA. For a 30 min reallocation of walking to sedentary behavior, the absolute effect size (absolute change in BP) for OPA is 11 times larger than for LTPA: -6.66 compared to +0.6 mmHg and reaches an effect size that is substantial and could be expected to decrease the risk of CVD by over 20% based on the known linear relationship between SBP and CVD. 

Although it is correct that all CIs include zero, there are notable differences in CIs across domains: CIs for LTPA point estimates are more balanced around the zero value while CIs for OPA point estimates are heavily tipped towards negative values (decreases) in BP. For the 30 min reallocation of walking to sedentary behavior mentioned above, the CIs for LTPA covers values between -2.66 to +3.85 and for OPA -16.19 to +2.88. These CIs are wide and thus indicate relative imprecise estimates that include zero, therefore the data are compatible with effects in either direction, especially for LTPA: the range of values within the CI interval for LTPA are nearly equally compatible with similar decreases or increases in SBP. In contrast, the majority of values within the CI for OPA are negative and negative values much larger than positive values, the data are therefore much more compatible with large decreases in SBP than (much smaller) increases in SBP. Taking these more detailed observations together, the overall results are clearly much more compatible with a detrimental effect of walking at work than not and also point in this direction much more than the results point into the direction of the smaller potential beneficial effect of LTPA. Despite a lack of statistical significance, these results are much more compatible with your hypothesis of the presence of the PA health paradox than with similar effects across domains. While your description of the results is not wrong, it implicitly relies too much on the single data point of statistical significance. Your description and especially your overall interpretation of the data should consider the entirety of information contained in CIs, not just the equivalence of one specific data point at the edge of the 95% CI interval that is equivalent to a p-value of 0.05. Instead of answering the narrow question, do 95% of all values lie above this one arbitrary point along the continuum of results within the CI, one needs to consider if the majority of points within the CI points to an increase or decrease in SBP. Several decades of scholarly work, textbooks of modern epidemiology (e.g. by Rothman, Greenland et al), and official statements of the American Statistical Association (2015) all encourage to base conclusions on effect estimation and making full use of the data included in CIs instead of relying primarily on statistical significance testing or its equivalent in the interpretation of CIs. 

In this spirit, I would suggest to represent all effect estimates in the tables with the same font and not to bold those data and CIs that do not include zero. The emphasis should be on results that are substantitive, e.g. point to a relevant change in the health risks that are known to be associated with your cardiovascular risk factor under study. For example, it is known that above about 115 mmHG, the relationship between SBP and CVD outcomes is monotone positive and virtually linear and that the risk of acute myocardial infarction (AMI) in a population increases by about 10% for each 2-3 mm Hg average increase in any population. Since AMI is a rather common disease, a 10% increase means thousands of people in DK and millions of people worldwide. Therefore, even a “small” 1 or 2 mmHG change in SPB that may be deemed irrelevant by clinicians for an individual patient is considered important in occupational and public health concerned with prevention of CVD in whole populations. From that perspective, researchers have considered even relative relative small changes of 1 mmHG or less as substantive and of public health significance. From that perspective, it is much more important to ask the question if 95% CI’s include values of that magnitude or more than the question if it includes zero values.

Response: We fully agree in the reviewer’s approach to interpretation of CIs and effect sizes and acknowledge that we have not communicated this clearly.

Action: We have changed the paragraph in the results section and, in accordance with the reviewer’s recommendations, emphasized the effect sizes. It now reads: “During leisure, the results indicated that less sedentary behaviour and more walking compared to the reference composition was associated with a lower SBP, while the results indicated an association with a higher SBP during work (Figure 2A and Table 3). Importantly, the size of the estimated differences in SBP differed markedly between the domains. For example, the absolute difference in SBP given 30 minutes less walking and 30 minutes more sedentary behaviour during work was 11 times larger than that during leisure (work: -6.7 [95% CI: -16.2, 2-9] mm Hg vs. leisure: 0.6 [-2.7, 3.8] mm Hg). The same pattern of opposite associations was evident for less sedentary behaviour and more HIPA during leisure and work. Although the CIs included zero, the majority of the values indicated a lower and higher SBP during leisure and work, respectively (e.g., 10 min, leisure: -0.7, 95% CI: -1.5, 0.2; Figure 2B and Table 4).” (p. 17-18, line: 341-351). 

Additionally, we have modified the last paragraph in the discussion of the SBP results. It now reads: “Our results indicated a 1.7 (95% CI: -0.8, 4.2) mm Hg higher SBP given 30 minutes less sedentary behaviour and 30 minutes more walking during work, and an 0.7 (95% CI: -2.6, 1.2) mm Hg lower SBP given the same time reallocation during leisure. Furthermore, 30 minutes less walking and 30 minutes more sedentary behaviour during work suggested a 6.7 (95% CI: -16.2, 2.9) mm Hg lower SBP. This difference is 11 times larger than that of the opposite reallocation during leisure (i.e., 30 min less sedentary behaviour and 30 min more walking: -0.7, 95% CI: -2.6, 1.2 mm Hg), and could be expected to reduce the risk of CVD-specific mortality by over 20% based on the known linear relationship between SBP and CVD (55, 56). Since even small changes in the population mean SBP can have important implications for CVD risk (i.e., affecting the prevalence of hypertension) (55-57), these findings could, potentially, have important implications in population-based prevention of CVD (45).” (p. 25-26, line: 488-499).

In the tables, bold is no longer used to highlight estimates where the CI does not include 0.

15) line 350-54 Results for WC: This description mentions results of “less sedentary behavior and more walking,” but not of “more sedentary behaviour and less walking” although the latter reallocation leads to substantially larger BP changes for both LTP and OPA. Similarly to SPB above, reallocation of 30 min of occupational walking to sedentary work led to a very substantial -5cm reduction of WC, five times as much than the reduction observed when the same amount of walking during leisure is reallocated to sedentary behavior. 

Response: The results of “more sedentary behaviour and less walking” is shown as the positive part of the x-axis in Figure 3. However, for clarity, we prefer to consistently describe the same direction of the time reallocations in the text. The asymmetry of the results mentioned by the reviewer is clearly seen in the figures, but we agree that this should be explicitly conveyed to the reader. 

Action: We have paraphrased the description of the results for WC. It now reads: “During both leisure and work, the results indicated less sedentary behaviour and more walking to be associated with a larger WC; however, the CIs included zero (Figure 3A and Table 3). In contrast, during leisure and work, less sedentary behaviour and more HIPA was associated with a smaller WC, although the estimates during work were small. Also, for work, the CIs included zero, but most values suggested a smaller WC (Figure 3B and Table 4). The estimated difference in WC given the time reallocations was not symmetric. For example, during work, the reallocation of 30 min walking to sedentary behaviour was associated with a 5 cm smaller WC (95% CI: -11.29, 1.03) compared to an estimated 1 cm larger WC given the opposite time reallocation. Additionally, the smaller WC (i.e., -5 cm) is about five times larger than the estimated difference observed for the corresponding time reallocation during leisure (i.e., -1 cm).” (p. 20-21, line: 369-379).

16) line 354-55 WC: the sentence “We found no association during work” is not backed up by the data. I would suggest to replace by a similar wording you used for HIPA and LDL in the next paragraph: “During work, the estimates followed the same pattern, but were even smaller …”

Response: We agree that the previous formulation was not backed up by data.

Action: We have omitted “We found no association during work (Figure 3B and Table 4).” and added: “In contrast, during leisure and work, less sedentary behaviour and more HIPA was associated with a smaller WC, although the estimates during work were small. Also, for work, the CIs included zero, but most values suggested a smaller WC (Figure 3B and Table 4).” (p. 20, line: 371-374).

17) line 357-362 Results for LDL, Table 4, row -1 min, column Work: CI does not include point estimate (data entry error?)

Response: Thank you for spotting this data entry error. 

Action: The numbers have been corrected to “ -0.01 (-0.03, 0.02)” (p. 19-20, Table 4, LDL-C, row -1 min, column work).

Discussion:

18) line 377-383 SBP: You state correctly “Although not statistically significant, these findings support…” the PA paradox. Since SBP is one of the most important global CVD risk factor, you may want to provide the reader with a more detailed interpretation of results that help the reader to understand why you do not dismiss results that are not statistically significant. See comment #14 above.

Response: We agree and have discussed the implications of the results in more depth in one of the subsequent paragraphs (i.e., p. 25-26, line: 488-499).

Action: Please see responses to comment 14.

19) line 391: replace “makes” by “make” (plural)

Response: Thanks for pointing out this error.

Action: We have changed “makes” to “make” (p. 24, line: 453). 

20) line 390-396: suggest to add the reference 18 to reference 14 each time mentioned here and add also consider to reference a couple of review papers that summarize the effect of BP and HR on CVD (some are cited in reference 18) if you have enough room for references. You may also mention the very simple explanation that cumulative exposure to higher BP and HR during work hours can be expected to increase CVD risk based on these positive associations between SPB and HR – it is a predictable outcome based on the well-established hemodynamic theory of arteriosclerosis (review paper by S Glagov 1988 “Hemodynamics and atherosclerosis” and/or M.J. Thubricar, Vascular mechanics and pathology, Springer, New York, 2007)

Response: We agree that the use of reference 18 is relevant. 

Action: Reference 18 (now reference 19) has been added in relation to the use of reference 14 (i.e., p. 24, line: 453, 455, and 458). 

21) line 401: insert “objective” before “device-based”

Response: We have chosen to consistently use “device-based measurements” instead of objective measurements since it has been questioned if the interpreted results from device-based measurements truly are “objective”.

Action: None.

22) line 426-33: In this paragraph you mention replacing sedentary activities with more walking at work may increase SBP by 1.7 mmHg and how this represents a substantially increased CVD risk on a population level. This makes totally sense and is in line with my earlier comments asking for stating this clearly. However, I think it is crucial to also point out the even higher prevention potential for replacing walking time at work by sedentary time by stating something like that (see comment #14 above) “While reallocation from reference to more sedentary LTPA behavior shows only small increases in BP between +0.21 and +0.60 mmHg, reduction of walking at work by 30 min and allocating this time to sitting at work results in rather substantial average decreases -6.66 mmHG. For a 30 min reallocation of walking to sedentary behavior, the absolute effect size (absolute change in BP) for OPA is infect 11 times larger than for LTPA: -6.66 compared to +0.6 mmHg and reaches an effect size that is substantial and could be expected to decrease the risk of CVD by over 20% based on the known linear relationship between SBP and CVD.” I think it is important to state this observation explicitly because the average reader who is has been reached by public health messages advocating a decrease of sitting at work will not pick this counter-intuitive statement that it may be actually much more advantageous to replace walking by sitting at work.

Response: The reviewer raises a good point, and we acknowledge that this message has not been clear and explicit in our wording.

Action: See action to detailed comment 14.

23) line 450-52: suggest to replace “total energy expenditure” by “diet” here because (a) energy expenditure itself is an independent CVD risk factor and (b) an integral part of relative aerobic workload the OPA measure that takes the important mismatch between physical work demands in terms of energy burned and workers cardiorespiratory fitness i.e. the capacity to burn that energy (V02max) into account. (c) There is little evidence that total energy expenditure is most important for WC or obesity, it seems that the source of energy (sugar vs protein or fat) and endocrine effects of sugar and thus the high sugar content of soda-drinks and processed food may instead be the determining factors for central obesity (see comprehensive in-depth review in book by Gary Taubes for the specific literature: “Good calories, bad calories: fats, carbs, and the controversial science of diet and health”). 

Response: We acknowledge the reviewer’s point.

Action: We have paraphrased so the sentence now reads: “The current findings also support that domain-specific characteristics of physical activity do not affect risk factors for which diet is most important (20, 63, 64).”

24) line 499-500: This statement (“i.e. only borderline during work”) is vague because it does not explain what “borderline” it refers to although I would assume that it is based on comparisons of statistical significance between PA domains instead of any minimum relevant effect size or effect estimation. The statement is problematic because the actually measured effects are different than this statement suggests: 50% higher for LTPA compared to OPA when adding 2 min of HIPA (0.02 vs 0.01 mmol/l), identical when adding or substracting 1 minute HIPA (0.01 mmol/l) and 50% higher for OPA (0.03 vs 0.2 mmol/l) when reducing HIPA by 2 min as shown in Table 4. 

Pointing out that only the LTPA findings are statistically significant does not summarize these results well. It looks more like consistent but small substantial effects overall regardless of direction and statistically significance testing. However, this assessment would be premature without consideration of how important any small LDL changes in the observed range maybe at the population level. You provide this info in your next paragraph: 30% for 1 mmol; this translates into 0.3% -0.9% lower IHD mortality, this seems to be still substantial on a population level although is much smaller than the examples given for SBP above. Your conclusion that the relationship between PA domains and LDL-C is unclear holds. However, this needs to be balanced against the potentially larger detrimental effects these reallocations may have with regard to SBP.

Response: We acknowledge that “borderline” is vague, can be misunderstood, and that the summary of the results could be improved in accordance with the reviewer’s interpretation. Furthermore, the relative differences between the domains (e.g., a 50% higher estimated difference in LDL-C for time reallocations during leisure compared to work) are indeed correct. However, for the time reallocations during work, the estimates are close to zero with relatively symmetrical CIs. It can therefore be problematic to make any strong interpretations of these results’ potential implications. We have therefore tried to balance the interpretation.

Action: The sentence now reads: “Furthermore, during both domains, less sedentary behaviour and more HIPA seemed to be associated with a lower LDL-C.” (p. 28, line: 564-565).

In addition, we have changed the following paragraph to reflect the potentially detrimental effects of the time reallocations on SBP that are larger than those on LDL-C. It now reads: “On a population-level, a 1 mmol/L lower non-HDL-C (i.e., total cholesterol minus HDL-C) has been reported to lower IHD-mortality by 30% (69). This translates to 0.3% lower IHD-mortality for every 0.01 mmol/L lower LDL-C. Therefore, even small improvements in LDL-C on a population-level like those observed in the current study, could, in combination with improvements in other modifiable risk factors (e.g., poor diet, high SBP, obesity, smoking, high alcohol consumption, and others), likely contribute to the prevention of incident IHD (70, 71). However, the potentially detrimental association between less sedentary behaviour and more HIPA during work and SBP should be kept in mind.” (p. 29, line: 573-581).

25) line 500 cited reference 47 by Honda 2014: It may be interesting to note that Honda reported (under fully adjusted model 3 in their Table 3) that 60 min of sedentary LTPA increases LDL by 0.77 mmol/L, while sedentary OPA, in contrast, decreases LDL by the -0.73 mmol/l. Similar paradoxical effects were shown for SBP (and less convincingly for WC) although the authors choose to totally ignore these findings because they were not statistically significant. However, a closer examination of confidence intervals (-1.78 to 0.31 for LDL) shows that their data are much more compatible with the PA paradox than not. I attach this table with my highlights here for your quick reference.

Response: We acknowledge that the contrasting findings in the study by Honda et al. were not clear from our previous wording.

Action: To better reflect the contrasting findings reported by Honda et al., this section now reads: “This disagrees with findings from three studies (52, 53, 65), where similar associations were reported for sedentary behaviour during leisure but not for work (except for the study by Honda et al. (49) where indications of opposite associations during leisure and work are reported).” (p. 28, line: 565-568). 

26) Line 510: “being overweight” is entered as a CVD risk factor. May be replacing it by “obesity” would be appropriate: If I recall it correctly, a closer look at the evidence suggests a bi-modal risk relationship between BMI with elevated risks for underweight and normal weight persons, lowest risk for overweight, and increasing risk for obese and above. 

Aside: Interestingly, in several Finnish cohort studies, BMI was not at all (HR=1.0) related to CVD and mortality outcomes in models that include occupational physical activity (and virtually all other known CVD risk factors). Hi BMI is also partly a function of high muscle mass which is related to the capacity to perform heavy labor and thus may be an intricate correlate of high OPA but with no independent effect on CVD once OPA is controlled for. Consequently, the literature on overweight and CVD is probably still inconclusive.’

Response: We acknowledge the reviewer’s point.

Action: We have replaced “being overweight” with “obesity” (p. 29, line: 578).

27) Line 519-20: From a primary prevention perspective it is paramount not to exclude 2/3 of the general population that has these potentially modifying factors – we want to know how this works out in real populations not a selected minority group of mostly healthy individuals who are already at the lowest risk for CVD, they are the least in need of additional research! It seems that you have data on persons with theses potentially confounding or effect modifying factors (based on the many exclusions you reported based on these factors). Could you rerun your analyses with everybody included and compare this with your current results as a sensitivity analysis? And thereafter investigate this further with stratified analyses by how much these factors may change relationships (in a supplement or in a separate paper)? See also comments and specific suggestions in comments #6 and #7 above.

Response: We have conducted sensitivity analyses based on a) all study participants, regardless of use of antihypertensives or cholesterol lowering drugs, and b) only study participants using antihypertensives, diuretics, or cholesterol lowering drugs. Please note that the exclusion criteria related to the use of specific medications did not exclude two thirds of the eligible study participants. As seen in Supporting information file S3, these sensitivity analyses included 804 and 146 study participants, respectively (compared to n=652 in the main analyses).

Action: Please see action in response to detailed comment 7. 

28) Line 527-529: While I agree that representativeness is not as important as validity (because what sense would it make to generalize findings of questionable validity?). I also agree that normal physiology needs to be studied in normal or healthy persons (but not necessarily young, or college-educated, or athletic etc. because what is normal will change during the life cycle – think of menopause and other normal changes). “Normal” is a quite difficult selection criterion as is “external”, after there are not really any people on earth that are not influenced by a myriad of “external” factors, medication is just one of the ones we actually can and should measure and thus take into account. 

Instead of controlling for these factors by either excluding participants or by adjustment in multivariate models, these factors and their influence on the exposure-disease relationships need to be actively investigated. Two basic steps have been suggested above: investigate the potential impact quantitatively by comparing changes in effect estimates (not statistical significance!) caused by adding this modifying factor to multivariate models and investigate multiplicative and additive interactions or (better) by comparing effect estimates from analyses stratified on categories of this potentially modifying factor. (More sophisticated mediation analysis may be needed to disentangle the different effects those factors may have but they are best reserved for prospective data with repeat measures). 

Exclusion of people with potentially effect modifying characteristics is not helping to clarify their potential impact and is actually detrimental for primary prevention efforts that need to address high risk populations that are often characterized by these very factors that led to the exclusion of otherwise eligible study participants. This is the main weakness of this paper but it could be addressed by adding the suggested sensitivity analyses.

The selection of analytic samples from general study populations for epidemiological investigation into the associations between ubiquitous PA exposures and highly prevalent CVD risk factors with a goal to solve occupational and public health problems associated with these risks, need to be different than samples selected for the study of physiological experiments, or sports performance, or clinical interventions designed for patients only. While I think that your study is a wonderful contribution, it would be best to strive for resolving these issues instead of excluding two thirds of eligible participants from further investigation (and thus basic knowledge needed to design interventions for them). I hope it is clear that I am not asking for restricting research to representative samples, I agree wholeheartedly with Rothman whom you cite on this topic, but he and I myself actually argue for non-representative analytic sampling strategies, including stratified analyses, that allow to make inferences of potentially modifying factors such as pre-existing CVD that have already been shown to exert strong modifying effects on CVD risk in other studies and are common in our societies and our aging workforce too. Again, sampling for such a study may include oversampling certain subgroups to make comparisons between subgroups more comparable (study design B in Rothman 2012 you cited). Thanks for citing his papers, it was a pleasure to reread them!

Response: We agree with the reviewer. The suggested stratified analysis has been conducted as a sensitivity analysis.

Action: Please see actions to detailed comment 7. 

29) line 537: limitations of exposure assessment by accelerator data: You may want to mention three additional important limitations here: 1) inability to consider the weight of materials, people, or tools handled (need for additional ergonomic assassments), 2) questionable implicit assumption that observed exposure during a short time window accurately reflects typical exposure (need to assess that separately by diary or survey or experts), 3) inability to assess past and cumulative overall exposure that may be most relevant for adverse chronic health effects such as CVD (need for repeat measurements over long study periods that is typically not feasible; need for assessing past exposure through detailed job histories based on self-report, administrative records, and expert assessments, e.g. a job exposure matrix, JEM). Occupational exposure assessment that solely relies on accelerometer-based snap-shots of current exposure is not necessarily more valid than self-reports or JEMs, and may in fact lead to massive exposure misclassification, especially if job tenure is short which is the case in many low SES jobs.

Response: We acknowledge that these are additional limitations that could be added to the manuscript. Also, we find that this strengthens the argument of not using objective device-based measurements as a term for a general exposure assessment. 

Action: We have changed the wording of the paragraph discussing limitations of the exposure assessment (p. 31). It now reads: “Firstly, the measurements do not capture the load in specific tasks such as heavy lifting, pushing, pulling, or awkward body positions (does not include measurements of the weight of materials, people, or tools handled), which are known to impose high physical demands, and therefore, could be important (13, 14). Secondly, common to all accelerometer-based measurements of physical behaviours, the measurements do not include the relative intensity of the physical activity. Thirdly, we do not know whether the measurement period accurately reflects the study participants’ typical physical activity level. Finally, we do not have data on job title, and on past or cumulative job exposure. These limitations imply a risk for misclassification of the exposure which, potentially, could lead to an underestimation of the health effects.” (p. 31, line: 627-636).

30) line 548-51 LDL-C: Comment: LDL-C appears to be indeed a strong and, as prospective cohort studies of OPA and CVD have shown, also a rather independent CVD risk factor that is therefore not a prime candidate for being a major mediator or confounder of the OPA -CVD relationship. However, this needs to be determined in prospective studies that also take the effects of widely-prescribed lipid-lowering drugs and pre-existing CVD into consideration.

Response: We thank the reviewer for this comment and agree that our results should trigger prospective studies investigating this, which would be of value to improve our understanding of these relationships. 

Action: None.

31) line 566-67: See comment # on “borderline” Although not the subject of any academic paper I am aware of, it is well-known among occupational health researchers, ergonomists, and manual workers themselves, that physically demanding work is only endurable if workers avoid as much as possible HIPAs that make them sweat or breathless during work (i.e. avoid the intensity that may produce training benefits in terms of cardiorespiratory fitness as it does during LTPA or athletic training) because working conditions typically do not allow for self-paced work with adequate recovery time and it is impossible for most workers to change out of sweat-drained wet clothes into dry clothes causing during work shifts. Therefore, larger sample sizes are not likely to change the noted limitation of few minutes of HIPA, it is an inherent characteristic of most real-world heavy labor.

Getting more accurate data instead will be dependent on implementing another form of composition analysis: improvement of all sources of exposure misclassification of OPA as outlined in comment #29 above: combining sensor-based methods with detailed job histories, JEMs, and other available data (e.g., administrative cumulative data on work hours, workdays, and leave time such as vacation, sick days, family leave, continuing education or retraining, unemployment, disability, retirement etc.) to construct more accurate exposure measures, and exposure measures that actually capture the relevant cumulative OPA exposures. Such improved imprecision of exposure assessment may help to reduce the conservative misclassification bias that plagues the literature and may be responsible for imprecision, underestimation of related chronic health consequences, and an overall inconclusive evidence base. Further, avoiding of categorization of continuous outcome measures artificially introduced by researchers during survey development or during data analysis, and replacing quantile categorization by categories that are demarcated by actual threshold effects in risk, is a much more promising and more feasible and cost-efficient approach for yielding more precise and also more valid health effect estimates. 

Response: We agree that “larger sample sizes are not likely to change the noted limitation of few minutes of HIPA”. However, the limitation we try to emphasise is the imprecision of the estimates. Given a larger sample size, the variation in our data set would likely be smaller, which would lead to more narrow CIs and hence more precise estimates. However, we also agree that combining device-based measurements with other types of exposure measurements could be an important approach to further this research area.

Action: We have paraphrased to avoid the word “borderline” (in line with response to comment #24). The paragraph now reads: “In general, the estimates were small, and the CIs were wide, in particular for the work-specific time reallocations. This is likely a consequence of the size of our study population, and the relatively small number of participants with a long duration of HIPA during work, which results in a large variation. A larger study population would likely result in less variation and thereby improved precision of the estimates, which could increase the confidence when interpreting the results.” (p. 32, line: 663-668).

32) line 569-72: I would also reference Peter Smith’s Canadian landmark cohort study on AMI here (“The relationship between occupational standing and sitting and incident heart disease over a 12-year period in Ontario, Canada,” Am J Epidem, 2018;187:27-33), which is also good example how JEM’s can be utilized for objective OPA assessment in very large samples that are typically beyond feasibility regarding accelerometer- or direct ergonomic observation.

Response: We agree that the suggested reference could be added, since it supports the statement.

Action: We have added the reference to the sentence “However, as previous studies and our results indicate (8, 12, 80), public health messages such as ‘sit less and move more’, may not be well suited for population groups that are highly physically active during work.” (p. 33, line: 673-675).

33) line 577: Can you be specific about the health outcomes and provide respective references here? 

Response: We have paraphrased to soften the statement.

Action: It now reads: “Currently, for several health outcomes it is still unclear how individuals with high occupational physical activity should best compensate during leisure.” (p. 33, line: 680-682).

34) line 584: I would suggest to cite here some of the existing evidence regarding null effects of LTPA (e.g., Krause N, Brand RJ, Arah OA, Kauhanen J, Occupational physical activity and 20-year incidence of acute myocardial infarction: results from the Kuopio Ischemic Heart Disease Risk Factor Study, Scand J Work Environ Health, 2015;41(2):124-139) and regarding even detrimental effects of LTPA (e.g., Eaton CB, Medalie JH, Flocke SA, Zyzanski SJ, Yaari S, Goldbourt U,Self-reported physical activity predicts long-term coronary heart disease and all-cause mortalities. Twenty- one-year follow-up of the Israeli Ischemic Heart Disease Study. Arch Fam Med 1995;4(4):323–329; more references can be found in PA guidelines talking about risks of LTPA), as well as specific literature regarding “overtraining effects in sports”?

Response: The paragraph has been paraphrased in relation to other comments, which made the suggested references redundant.

Action: None.

35) line 587: Reference 73(Korshøj et al. 2015) does not support the introduction of aerobic exercise during work. This RCT showed average increases of systolic blood pressure by 3.6 mm Hg (95% CI 1.1-6.0) relative the control group. It is important to cite this study but as a warning regarding unintended consequences of such a program instead as confirmation for this approach.

Response: This is true, and this reference deserves more attention. However, in a larger perspective considering all monitored effects, the most important is that a positive change was seen for a number of clinically recognized risk factors. That is, although the mean SBP increased following the intervention, we consider the benefits of the exercise intervention to outweigh the potential harms.

Action: The potential of unintended consequences has been emphasised. The section now reads: “One potential alternative is workplace-based initiatives, such as aerobic exercise during work hours. Although such interventions may have unintended negative health effects such as increased SBP (81), they can improve cardiorespiratory fitness, workability, and health (81-83).” (p. 33, line: 682-685).

36) line 587-99: The point is not to consider OPA in PA, that is already being done implicitly by counting any PA towards recommended PA goals. The pressing issue for a revision of PA guidelines is instead to acknowledge the PA health paradox and the need for more targeted and safe PA recommendations that take potential differential effects of OPA and LTPA and baseline CVD health status into account.

Response: We acknowledge the reviewer’s point and have paraphrased. 

Action: The sentence now reads: “It is, therefore, highly important to take the potentially contrasting health effects of leisure time- and occupational physical activity into account in physical activity recommendations for adults.” (p. 33, line: 685-687).

37) line 596: see comment #31 above re “larger sample size”. Maybe avoidance of unnecessary misclassification is worth mentioning here?

Response: We agree that a short sentence about how the exposure assessment may be improved (with less misclassification) in future studies could fit nicely here. 

Action: In continuation to detailed comment 29 and 31, we have added “Combining device-based measurements with data on previous job titles, job exposure matrices, routinely collected administrative data (e.g., periods of sick leave periods, retirement), or questionnaire data to improve the exposure assessment and minimise misclassification could be a fruitful avenue for future studies.” (p. 34, line: 697-701).

38) line 602-609: This overall conclusion summarizes the main findings well and is backed up by the data. Given the discussion of WC and LDL_C results above, it may be appropriate to replace “No difference between domains ..” by “No consistent differences ….”

Response: We see the reviewer’s point and have paraphrased in line with the suggestion.

Action: It now reads: “In contrast, no consistent differences between domains were observed for WC and LDL-C.” (p. 34, line: 707-708).

References

1. Skotte J, Korshoj M, Kristiansen J, Hanisch C, Holtermann A. Detection of physical activity types using triaxial accelerometers. Journal of physical activity & health. 2014;11(1):76-84.

2. Unger T, Borghi C, Charchar F, Khan NA, Poulter NR, Prabhakaran D, et al. 2020 International Society of Hypertension Global Hypertension Practice Guidelines. Hypertension. 2020;75(6):1334-57.

---

## [Decision Letter · Decision Letter 1]

7 Feb 2022

PONE-D-21-00560R1The physical activity health paradox and risk factors for cardiovascular disease: a cross-sectional compositional data analysis in the Copenhagen City Heart StudyPLOS ONE

Dear Dr. Johansson,

Thank you for submitting your manuscript to PLOS ONE. After careful consideration, we feel that it has merit but does not fully meet PLOS ONE’s publication criteria as it currently stands. Therefore, we invite you to submit a revised version of the manuscript that addresses the points raised during the review process.

We look forward to receiving your revised manuscript.

Kind regards,

Gianluigi Savarese

Academic Editor

PLOS ONE

Journal Requirements:

Additional Editor Comments:

Minor comments:

The Background is definitely too long and reads as a review. The Authors are invited to shorten this section. Most of what explained here belongs to the discussion.

There were some missing data. How were they handled?

Reviewers' comments:

Reviewer's Responses to Questions

**Comments to the Author**

1. If the authors have adequately addressed your comments raised in a previous round of review and you feel that this manuscript is now acceptable for publication, you may indicate that here to bypass the “Comments to the Author” section, enter your conflict of interest statement in the “Confidential to Editor” section, and submit your "Accept" recommendation.

Reviewer #1: All comments have been addressed

2. Is the manuscript technically sound, and do the data support the conclusions?

Reviewer #1: Yes

3. Has the statistical analysis been performed appropriately and rigorously? 

Reviewer #1: Yes

4. Have the authors made all data underlying the findings in their manuscript fully available?

Reviewer #1: Yes

5. Is the manuscript presented in an intelligible fashion and written in standard English?

Reviewer #1: Yes

6. Review Comments to the Author

Reviewer #1: I am willing to accept the manuscript. I will just recommend the authors to go through the paper and review carefully grammar and spelling to make sure there are no mistypes or something.

Otherwise, congratulations on this good paper.

7. PLOS authors have the option to publish the peer review history of their article (what does this mean?). If published, this will include your full peer review and any attached files.

Reviewer #1: No

---

## [Author Response · Author response to Decision Letter 1]

10 Mar 2022

Response to reviewers’ comments

PONE-D-21-00560

The physical activity health paradox and risk factors for cardiovascular disease: a cross-sectional compositional data analysis in the Copenhagen City Heart Study

Dear Dr. Gianluigi Savarese,

We would like to acknowledge you and the reviewer for taking your time to assess our manuscript and providing valuable feedback. Re the Journal requirements, we have checked the reference list to make sure it is complete and correct. A point-by-point response can be found below. Changes made in the manuscript have been highlighted using the track-changes function in Word. 

With kind regards on behalf of all authors,

Melker S. Johansson

Additional Editor Comments:

Minor comments:

1. The Background is definitely too long and reads as a review. The Authors are invited to shorten this section. Most of what explained here belongs to the discussion.

Response: We acknowledge that the introduction was too long. 

Action: The introduction has been shortened substantially from 636 words to 352 (p. 3-4, line: 55-84.

2. There were some missing data. How were they handled?

Response: We interpret the editor’s comment to relate to the missing values of some of the covariates included in the adjusted regression models. We acknowledge that the sentence given in the legends of Table 3 and Table 4 was not clear and have therefore paraphrased.

Action: The sentence in the legend of Table 3 and Table 4 now reads: “69 observations were not included in the adjusted models due to missing values in some covariates.” (p. 18 and 19). In addition, we have added the following sentence in the Methods section: “Observations with missing values in the covariates were not included in the adjusted models (n=69).” (p.11, line: 247-248).

Reviewers' comments:

Reviewer 1

I am willing to accept the manuscript. I will just recommend the authors to go through the paper and review carefully grammar and spelling to make sure there are no mistypes or something. Otherwise, congratulations on this good paper.

Response: We thank the reviewer for the comments, which have helped us improve the manuscript. Finally, we have reviewed the manuscript for grammatical and spelling errors and typos.

---

## [Editor Report · Decision Letter 2]

11 Apr 2022

The physical activity health paradox and risk factors for cardiovascular disease: a cross-sectional compositional data analysis in the Copenhagen City Heart Study

PONE-D-21-00560R2

Dear Dr. Johansson,

We’re pleased to inform you that your manuscript has been judged scientifically suitable for publication and will be formally accepted for publication once it meets all outstanding technical requirements.

Kind regards,

Gianluigi Savarese

Academic Editor

PLOS ONE

---

## [Editor Report · Acceptance letter]

14 Apr 2022

PONE-D-21-00560R2 

The physical activity health paradox and risk factors for cardiovascular disease: a cross-sectional compositional data analysis in the Copenhagen City Heart Study 

Dear Dr. Johansson:

I'm pleased to inform you that your manuscript has been deemed suitable for publication in PLOS ONE. Congratulations! Your manuscript is now with our production department. 

Kind regards, 

on behalf of

Dr. Gianluigi Savarese 

Academic Editor

PLOS ONE